**Data Availability Statement:** https://figshare.com/projects/Scientific_Quality_of_COVID-19_and_SARS_CoV-2_Publications_in_the_Highest_

# Scientific quality of COVID-19 and SARS CoV-2 publications in the highest impact medical journals during the early phase of the pandemic: A case control study

**Marko Zdravkovic**[1⊙], **Joana Berger-Estilita**[2⊙], **Bogdan Zdravkovic**[1], **David Berger**[3]*

1 Department of Anaesthesiology, Intensive Care and Pain Management, University Medical Centre Maribor, Maribor, Slovenia, 2 Department of Anaesthesiology and Pain Medicine, Inselspital, Bern University Hospital, University of Bern, Bern, Switzerland, 3 Department of Intensive Care Medicine, Inselspital, Bern University Hospital, University of Bern, Bern, Switzerland

⊙ These authors contributed equally to this work.
* david.berger@insel.ch

## Abstract

### Background

A debate about the scientific quality of COVID-19 themed research has emerged. We explored whether the quality of evidence of COVID-19 publications is lower when compared to nonCOVID-19 publications in the three highest ranked scientific medical journals.

### Methods

We searched the PubMed Database from March 12 to April 12, 2020 and identified 559 publications in the New England Journal of Medicine, the Journal of the American Medical Association, and The Lancet which were divided into COVID-19 (cases, n = 204) and nonCOVID-19 (controls, n = 355) associated content. After exclusion of secondary, unauthored, response letters and non-matching article types, 155 COVID-19 publications (including 13 original articles) and 130 nonCOVID-19 publications (including 52 original articles) were included in the comparative analysis. The hierarchical level of evidence was determined for each publication included and compared between cases and controls as the main outcome. A quantitative scoring of quality was carried out for the subgroup of original articles. The numbers of authors and citation rates were also compared between groups.

### Results

The 130 nonCOVID-19 publications were associated with higher levels of evidence on the level of evidence pyramid, with a strong association measure (Cramer's V: 0.452, P <0.001). The 155 COVID-19 publications were 186-fold more likely to be of lower evidence (95% confidence interval [CI] for odds ratio, 7.0–47; P <0.001). The quantitative quality score (maximum possible score, 28) was significantly different in favor of nonCOVID-19 (mean difference, 11.1; 95% CI, 8.5–13.7; P <0.001). There was a significant difference in

Impact_Medical_Journals_during_the_Early_
Phase_of_the_Pandemic_A_Case-Control_Study/
86027.

**Funding:** The author(s) received no specific
funding for this work.

**Competing interests:** Marko Zdravkovic, Bogdan
Zdravkovic and Joana Berger-Estilita have declared
that no competing interests exist. David Berger has
read the journal's policy and the authors of this
manuscript have the following competing interests:
The Department of Intensive Care Medicine at
Inselspital has, or has had in the past, research
contracts with Abionic SA, AVA AG, CSEM SA,
Cube Dx GmbH, Cyto Sorbents Europe GmbH,
Edwards Lifesciences LLC, GE Healthcare, ImaCor
Inc., MedImmune LLC, Orion Corporation,
Phagenesis Ltd. and research & development/
consulting contracts with Edwards Lifesciences
LLC, Nestec SA, Wyss Zurich. The money was paid
into a departmental fund; Dr Berger received no
personal financial gain. The Department of
Intensive Care Medicine has received unrestricted
educational grants from the following organizations
for organizing a quarterly postgraduate educational
symposium, the Berner Forum for Intensive Care
(until 2015): Abbott AG, Anandic Medical Systems,
Astellas, AstraZeneca, Bard Medica SA, Baxter, B |
Braun, CSL Behring, Covidien, Fresenius Kabi,
GSK, Lilly, Maquet, MSD, Novartis, Nycomed,
Orion Pharma, Pfizer, Pierre Fabre Pharma AG
(formerly known as RobaPharm). The Department
of Intensive Care Medicine has received
unrestricted educational grants from the following
organizations for organizing bi-annual
postgraduate courses in the fields of critical care
ultrasound, management of ECMO and mechanical
ventilation: Abbott AG, Anandic Medical Systems,
Bard Medica SA., Bracco, Dräger Schweiz AG,
Edwards Lifesciences AG, Fresenius Kabi
(Schweiz) AG, Getinge Group Maquet AG, Hamilton
Medical AG, Pierre Fabre Pharma AG (formerly
known as RobaPharm), PanGas AG Healthcare,
Pfizer AG, Orion Pharma, Teleflex Medical GmbH.
here are no patents, products in development or
marketed products associated with this research to
declare. This does not alter our adherence to PLOS
ONE policies on sharing data and materials.

**Abbreviations:** CI, confidence interval; COVID-19,
coronavirus disease 2019; QUALSYST, Standard
quality assessment criteria for evaluating primary
research papers from a variety of fields; SARS-
CoV-2, severe acute respiratory syndrome
coronavirus 2.

the early citation rate of the original articles that favored the COVID-19 original articles
(median [interquartile range], 45 [30–244] vs. 2 [1–4] citations; $P < 0.001$).

## Conclusions

We conclude that the quality of COVID-19 publications in the three highest ranked scientific
medical journals is below the quality average of these journals. These findings need to be
verified at a later stage of the pandemic.

## Introduction

Coronavirus disease 2019 (COVID-19) is caused by severe acute respiratory syndrome corona-
virus 2 (SARS CoV-2), and it is a rapidly spreading pandemic that is putting extraordinary
stress on healthcare systems across the globe (For simplicity, we will use COVID-19 in refer-
ence to both the virus and the disease). While everyone waits for a breakthrough of a specific
COVID-19 therapy and an effective vaccine, scientists are redirecting their efforts into
COVID-19–themed research to build up our knowledge of this new disease [1]. A search for
"COVID-19 or SARS-CoV2" in the PubMed database revealed 4,670 publications between
January 1, 2020, and April 12, 2020. This need to publish COVID-19–related findings has been
supported by many Ethical Committees, grant providers, and journal editors, who have 'fast-
tracked' COVID-19 publications so that they can be processed at record speed [2–4]. However,
concerns are emerging that scientific standards are not being met.

The first report of COVID-19 transmission in asymptomatic individuals [5] was later con-
sidered to have been flawed, because the patient showed symptoms at the time of transmission
[6]. A similar example occurred in *The Lancet*, whereby the authors retracted a publication
after admitting irregularities on the first-hand account of the front-line experience of two Chi-
nese nurses [7]. While our article was under review, two major analyses on the use of hydroxy-
chloroquine and cardiovascular mortality associated with COVID-19 were retracted in the
Lancet [8] and the New England Journal of Medicine [9] because source data could not be
verified.

Such situations raise concerns as to the quality of the data, the conclusions presented by the
authors, and the peer review by the editors, due to the pressure to publish highly coveted infor-
mation on COVID-19. The urgency of the outbreak suddenly appears to legitimize key limita-
tions of studies, such as small sample sizes, lack of randomization or blinding, and unvalidated
surrogate endpoints [10, 11].

While clinicians and the public long for effective treatments, a debate about the quality of
this surge of research and the potential violations of scientific rigor has emerged [10, 12, 13].
Despite this massive publication effort, current guidelines remain without any recommenda-
tions on core topics for patient management and care [14, 15]. The combination of clinical
urgency, weak evidence, pre-print publications without prior peer review [16], and public
pressure [17] might lead to inappropriate public health actions and incorrect translation into
clinical practice [18], with the potential for worrying breaches in patient safety [19]. A further
concern is the inflation of publication metrics, particularly in terms of journal impact factors.
Citation-based metrics are used by researchers to maximize the citation potential of their arti-
cles [20]. The expectation of a high citation rate might be used by journals to publish papers of
questionable scientific value on 'trendy' topics [21].

To date, the quality of COVID-19 publications in the top three general medical journals by impact factor (i. e. the New England Journal of Medicine, The Lancet and The Journal of the American Medical Association, represented by an impact factor > 50 for all) has not been formally assessed. We hypothesized that the quality of recent publications on COVID-19 in the three most influential medical journals is lower than for nonCOVID-19 articles published during the same time period. We also determined the early research impact of COVID-19 original articles *versus* nonCOVID-19 original articles.

## Materials and methods

This report follows the applicable STROBE guidelines for case-control studies.

### Publication selection and identification of cases and controls

For the time period of March 12 to April 12, 2020 (i.e., during the early outbreak phase of the COVID-19 pandemic), we identified all of the publications from the top three general medical journals by impact factor (the New England Journal of Medicine (NEJM), the Journal of the American Medical Association (JAMA), and The Lancet). We conducted a PubMed database search on April 17, 2020, using the following search string: *((("The New England journal of medicine"[Journal]) OR "Lancet (London, England)"[Journal]) OR "JAMA"[Journal]) AND ("2020/03/12"[Date—Publication]: "2020/04/12"[Date—Publication]).* The resulting publications were stratified into COVID-19–related and nonCOVID-19–related. We matched the nonCOVID-19 publications with COVID-19 publications according to article types within each journal, with the exclusion of nonmatching article types. Secondary studies, correspondence letters on previously published articles, unauthored publications, and specific article types not matching any of the six categories on the levels of the evidence pyramid [22–24] (e.g., infographic, erratum) were excluded (Fig 1).

### Multi-step design

We performed a multi-step 360-degree assessment of the studies. It consisted of their classification according to level of evidence for a quantitative appraisal of their methodological quality using a validated tool, and a narrative analysis of the strengths and weaknesses of the COVID-19 publications, as is often used in social sciences [25]. Early citation frequencies of the original articles was determined.

### Levels of evidence

All of the publications included were assessed for number of authors and level of evidence. We used the Oxford Quality Rating Scheme for Studies and Other Evidence [22] to categorize the level of evidence, as adjusted to include animal and *in-vitro* research [23, 24]. The highest level is attributed to research as randomized trials, followed by nonrandomized controlled studies and cohort trials. The lower levels are represented by descriptive studies, expert opinion, and animal or *in-vitro* research, commonly represented in the form of a pyramid [22, 23, 26]. For secondary analysis, we split the six levels of evidence into the upper and lower halves, which reflected higher (i.e., 1–3) and lower (i.e., 4–6) levels of evidence, respectively. The number of authors per publication was counted manually.

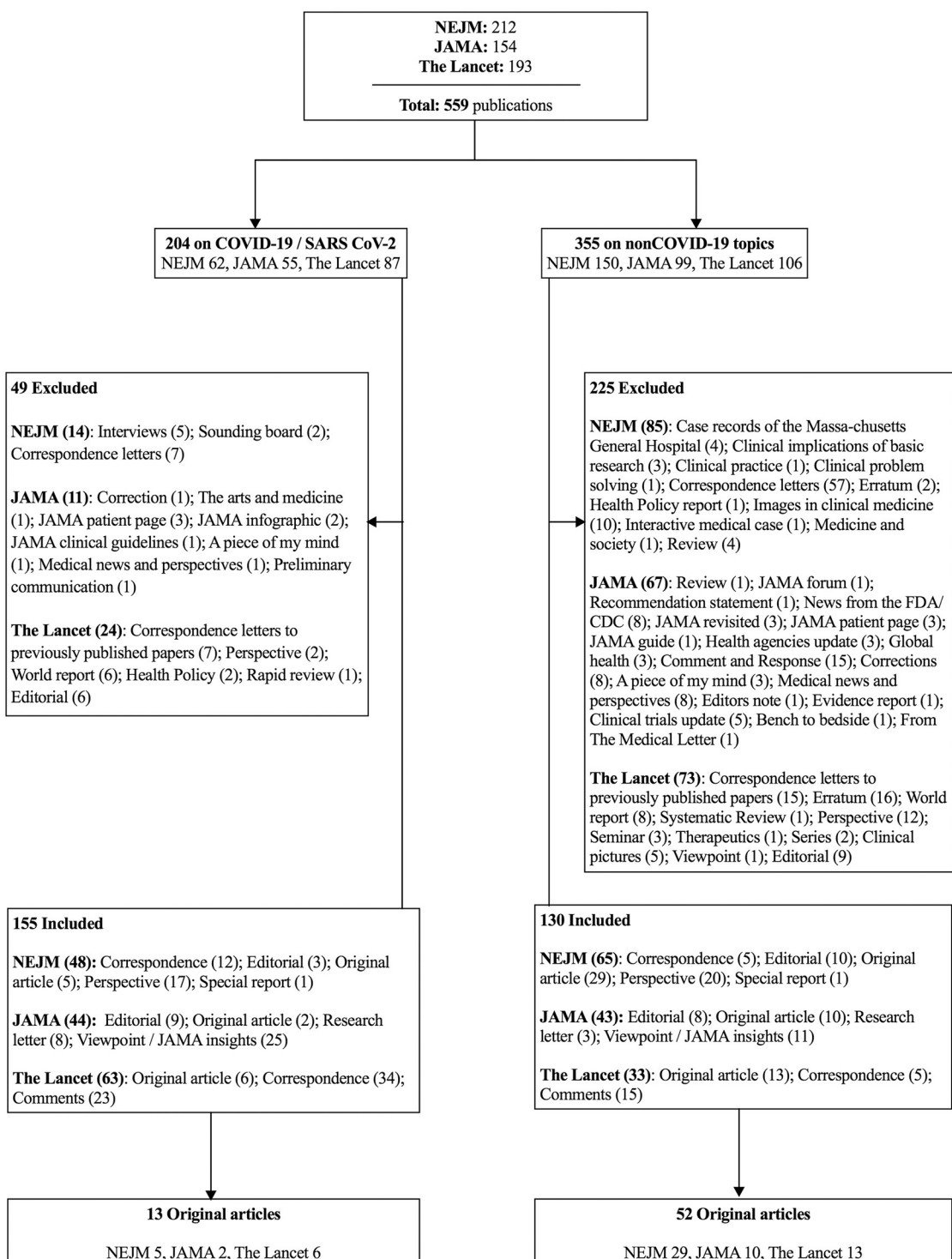

**Fig 1. Flow chart of the processing of the publications included in this study.** The article types in the NEJM are grouped (by the publisher) into *Original Research* (Research Articles and Special Articles for research on economics, ethics, law and health care systems), *Clinical Cases* (Brief Reports and Clinical Problem Solving), *Review Articles* (Clinical Practice Review or Other Reviews), *Commentaries* (Editorials, Perspectives, Clinical Implications of Basic Research, Letters to the Editor, Images and Videos in Clinical Medicine), and *other articles* (Special Reports, Policy Reports, Sounding Board, Medicine and Society and Case Records of the Massachusetts General Hospital). The JAMA articles are grouped by the publisher into *Research* (Original Investigation, Clinical Trials, Caring for the Critically Ill Patient, Meta-Analysis, Brief Reports and Research letters), *Clinical Review and Education* (Systematic Reviews, Advances in Diagnosis and Treatment, Narrative Reviews, Special Communications, Clinical Challenges,

Diagnostic Test Interpretation, Clinical Evidence Synopsis), *Opinion* (Viewpoints), *Humanities* (The Arts and Medicine, A Piece of My Mind, Poetry) and *Correspondence* (Letters to the Editor). The Lancet's articles are grouped into a *Red Section* (Articles and Clinical Pictures), a *Blue Section* (Comments, World Reports, Perspectives, Obituaries, Correspondence, Adverse Drug Reactions and Department of Error) and a *Green Section* (Seminars, Reviews, Therapeutics, Series, Hypothesis, Other Departments and Commissions).

### Quantitative appraisal using the "Standard quality assessment criteria for evaluating primary research papers from a variety of fields (QUALSYST)"

After the hierarchical grading of included publications, the original articles (i.e., published as 'original research articles' in each of the journals; Fig 1) were defined for further in-depth analysis using the study quality checklist proposed by Kmet et al. [27]. This checklist is consistent with the recommendations from the Center for Reviews and Dissemination [28, 29]. Four authors in pairs (MZ–DB, JBE–BZ; each pair assessing one half of the publications) independently assessed the original articles on 14 quality criteria (**see** S1 File). The 14 items covered the research question, design, measures to reduce bias, and data reporting and interpretation, and these were scored according to the degree to which each specific criterion was met ("yes" = 2; "partial" = 1; "no" = 0; "not applicable" = n/a) with the help of a prespecified manual [27]. The total score ranged from 0 to 28. The summary percentage scores were calculated for each original article by summing the total score obtained across the applicable items and dividing by the total possible score (i.e., 28 –[number of "n/a" $\times$ 2] $\times$100). Disagreements between the reviewers (defined as >2 difference in the total score, or >10% difference in the summary percentage scores), were resolved through one round of discussion between each 2-author pair.

### Narrative analysis of COVID-19 original articles

The COVID-19 original research articles (n = 13) were assessed in narrative form to report on their major weaknesses, potential conflicts of interest, and likely influence on further research and clinical practice.

### Citation frequencies

The early citation frequencies were tracked every 5 days from April 25th to May 25th 2020 for all of the original scientific articles through GoogleScholar [30], to determine how strongly these COVID-19 original articles had impacted upon further publications, in comparison to the nonCOVID-19 original articles. A comparison to an original article set in the same time frame of 2019 was done. Citations per month were calculated to reduce lead time bias. The Google scholar search engine has been shown to reliably identify the most highly-cited academic documents [31].

### Statistical analysis

The distributions of the COVID-19 and nonCOVID-19 publications on the levels of evidence pyramid were analyzed using Pearson's Chi-squared statistics and Cramer's V as the measure of strength of association (weak: >0.05; moderate: >0.10; strong: >0.15; very strong: >0.25) [32]. Further effect size estimations were performed on two by two contingency tables (split by level of evidence into high and low quality groups) and are reported as odds ratios with 95% confidence intervals (CI).

The retrospectively calculated sample size for the summary percentage scores [27] to detect a 20% change from 90 (nonCOVID-19) to 72 (COVID-19), with 4:1 allocation (52:13 original

articles, respectively) on a t-test, with a standard deviation of 15, 85% power, and 0.05 alpha, was 8 original articles [33, 34]. Thus, we deemed our collected data sufficient.

We also planned for a secondary analysis if the comparison above resulted in a significant difference (defined as P <0.05) in the mean percentage scores between the COVID-19 and nonCOVID-19 original articles. The secondary analysis aimed to compare the 2:1 allocation of nonCOVID-19:COVID-19 original articles, for which the allocation was carried out with the 26 original articles with the lowest overall percentage scores in the nonCOVID-19 group *versus* all of the 13 original articles in the COVID-19 group. The threshold p-value for significance was set at P <0.025, to adjust for multiple testing.

Assessment of the original articles' quality is reported as a two-reviewer mean score (95% CI) and was analyzed using Welch's t-tests. Hedges's *g* was used as the effect size measure based on a standardized mean difference [35] (small: $d = 0.20$; medium: $d = 0.50$; large: $d = 0.80$; very large: $d = 1.20$; huge: $d > 2.00$) [36, 37]. To confirm the reliability of the scoring, Cronbach's alpha was calculated for the total score and the summary percentage score (internal consistency), and the Intraclass Correlation Coefficient with absolute agreement for the inter-rater reliability. The percentage agreement between the two reviewers was also calculated for each individual item (**see** S2 File).

The data distributions were tested for normality with Kolmogorov-Smirnov tests, and are reported accordingly. Tests between two groups were done with Mann-Withney tests, between multiple groups with Kurskal-Wallis test. Significance was set at P <0.05 or adjusted for multiple testing. All of the tests were two-tailed. The statistical analysis was performed using SPSS Statistics 20 (IBM Inc., Armonk, NY, USA) and Prism 8 (GraphPad Software, San Diego, CA, USA).

## Results

Out of 559 publication entries on PubMed for the selected journals, 155 publications on COVID-19 and 130 publications on other (nonCOVID-19) topics were included in the level of evidence analysis. The subsequent analysis of quality was performed on 13 COVID-19 original articles in comparison with 52 nonCOVID-19 original articles (Fig 1).

### Levels of evidence and number of authors

The nonCOVID-19 publications were associated with higher quality on the level of evidence pyramid (P <0.001; Chi squared), with a strong association measure (Cramer's V: 0.452, Table 1). When comparing the higher evidence group to the lower evidence group, the COVID-19 publications were 18-fold more likely (i.e., odds ratio) to be in the lower evidence group (95% CI: 7.0–47; P <0.001). When comparing only the original articles on the levels of evidence pyramid (Table 2), the nonCOVID-19 publications were also associated with higher

**Table 1. Frequency distribution of the publications included on the levels of evidence pyramid [23, 24].**

| Study design | Level | Group | COVID-19 (n = 155) [n (%)] | nonCOVID-19 (n = 130) [n (%)] |
|---|---|---|---|---|
| Randomized controlled trial | 1 | Higher level of evidence | 1 (0.6) | 38 (29.2) |
| Well-designed controlled trial without randomization; prospective comparative cohort trial | 2 | | 0 (0) | 2 (1.5) |
| Case-control study; retrospective cohort study | 3 | | 4 (2.6) | 9 (6.9) |
| Case series without or with intervention; cross-sectional study | 4 | Lower level of evidence | 19 (12.3) | 10 (7.7) |
| Opinion papers; case reports | 5 | | 129 (83.2) | 69 (53.1) |
| Animal or *in-vitro* research | 6 | | 2 (1.3) | 2 (1.5) |

**Table 2. Frequency distribution of the original articles on the levels of evidence pyramid [23, 24].**

| Study design | Level | Group | COVID-19 (n = 13) [n (%)] | nonCOVID-19 (n = 52) [n (%)] |
|---|---|---|---|---|
| Randomized controlled trial | 1 | Higher level of evidence | 1 (7.7) | 38 (73.1) |
| Well-designed controlled trial without randomization; prospective comparative cohort trial | 2 | | 0 (0) | 1 (1.9) |
| Case-control study; retrospective cohort study | 3 | | 2 (15.4) | 7 (13.5) |
| Case series without or with intervention; cross-sectional study | 4 | Lower level of evidence | 9 (69.2) | 6 (11.5) |
| Opinion papers; case reports | 5 | | 1 (7.7) | 0 (0) |
| Animal or *in-vitro* research | 6 | | 0 (0) | 0 (0) |

quality (P <0.001; Chi squared), with a strong association measure (Cramer's V: 0.641, Table 2). When comparing the higher evidence group to the lower evidence group, the COVID-19 original articles were 26-fold more likely (i.e., odds ratio) to be in the lower evidence group (95% CI: 5.4–120; P <0.001).

Numbers of authors were similar between groups (median [interquartile range]: 3 [2–6.5] *versus* 3 [2–13.5]; P = 0.394; Mann-Whitney). In an *a posteriori* subgroup analysis in the lower evidence group (adjusted threshold p-value as P <0.017), there were significantly more authors in the COVID-19 publications (median [interquartile range]: 3 [2–6]) than in the non-COVID-19 publications (median: 2 [1–3]) (P <0.001; Mann-Whitney). Obvious outliers were a NEJM case report [38] with 35 authors, an opinion correspondence piece in The Lancet [39] with 29 authors, and a comment piece in The Lancet with 77 authors in a coalition [40].

## Quantitative appraisal

Due to >2 difference in the total scores, or >10% difference in the summary percentage scores, the reviewer pairs discussed 8 (of 32) and 12 (of 33), respectively, of the original articles after the individual scoring. The internal consistency reliability of the total score was 0.987, and of the summary percentage score was 0.964 (Cronbach's alphas) for the reviewer pair MZ–DB, and 0.988 and 0.928, respectively, for the reviewer pair JBE–BZ (P < 0.001, for all). The inter-rater reliabilities of the total scores was 0.975, and the summary percentage score was 0.930 (Intraclass Correlation Coefficient, absolute agreement) for pair MZ–DB, and 0.974 and 0.860, respectively, for pair JBE–BZ (Intraclass Correlation Coefficient, absolute agreement) (P < 0.001, for all).

The mean total scores in the COVID-19 and nonCOVID-19 groups were 12.6 (95% CI 10.1–15.1) and 23.7 (95% CI 22.9–24.6) respectively (Fig 2A), and the mean summary percentage scores were 71.8% (95% CI 62.4–81.1) and 91.1% (95% CI 89.0–93.2), respectively (Fig 2C). The mean total score and the mean summary percentage scores were significantly different between the groups, favoring the nonCOVID-19 original articles (P <0.001, for both; Welch's t-test; Hedges' g = 3.37, 2.02, respectively). For the total scores, the difference between the means was 11.1 (95% CI 8.5–13.7; P <0.001), and for the summary percentage scores, 19.3 (95% CI 9.8–28.8; P <0.001). Also, in the secondary analysis, when the COVID-19 original articles were compared to the lower quality half of the nonCOVID-19 original articles (i.e., the 26 scoring lower instead of all 52), the differences in the mean total scores (Fig 2B; 12.6 [95% CI 10.1–15.1] *vs* 21.4 [95% CI 20.4.1–22.3] points, respectively; P = 0.008; Welch's t-test; Hedges' g = 2.86) and the mean summary percentage scores (Fig 2D; 71.8% [95% CI 62.4–81.1] *vs* 85.6% [95% CI 82.8–88.5], respectively; P <0.001; Welch's t-test; Hedges' g = 1.31) were significant. For this secondary analysis, the threshold P value for significance was set at p = 0.025.

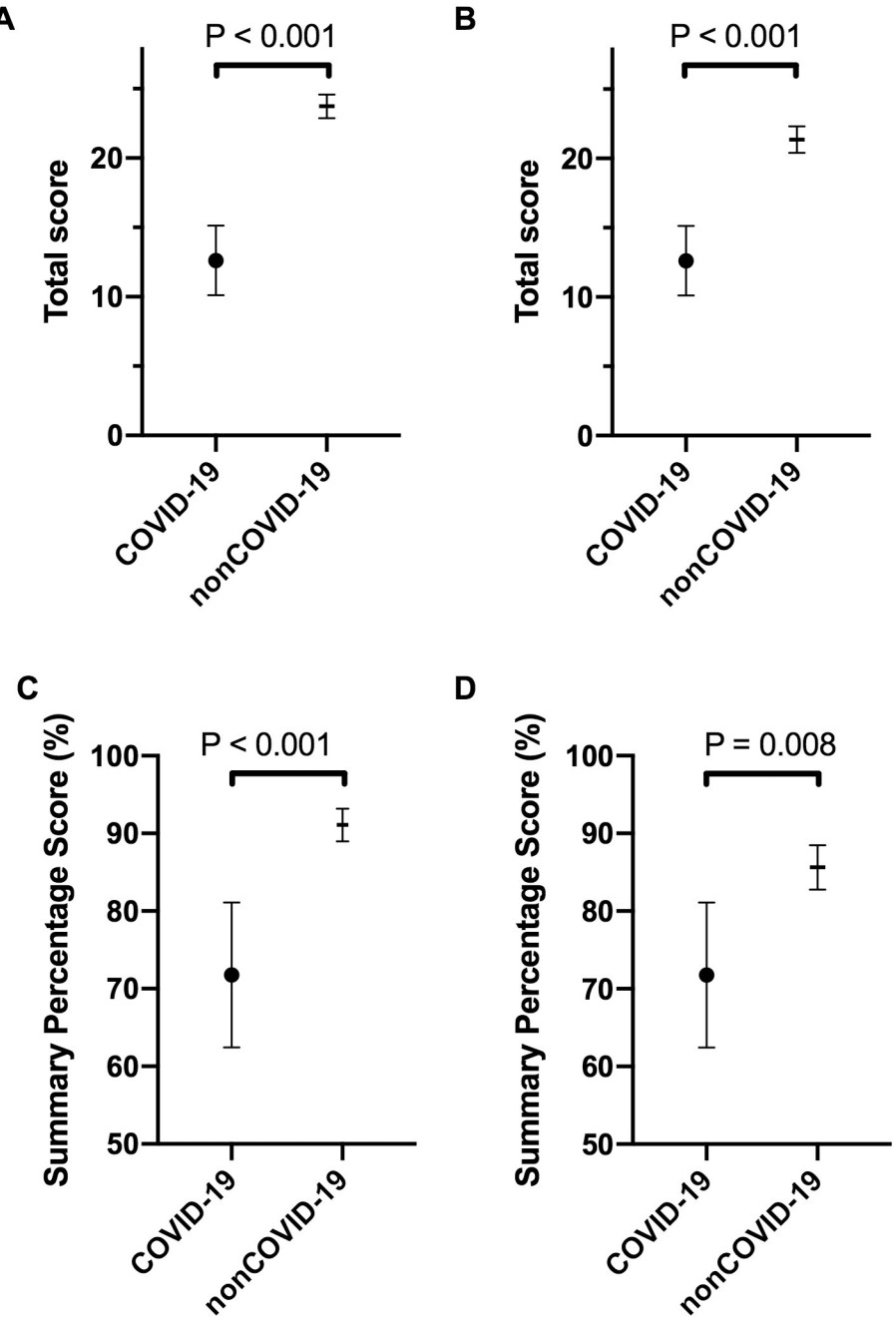

**Fig 2. Quantitative appraisal of the quality of the COVID-19 versus nonCOVID-19 original articles.** The "Standard quality assessment criteria for evaluating primary research papers from a variety of fields"[25] was used, for a maximum total score of 28. (**A, C**) Primary analysis for mean total scores (A) and mean summary percentage scores (C) for all COVID-19 (n = 13) and nonCOVID-19 (n = 52) original articles. (**B, D**) Secondary analysis for mean total scores (B) and mean summary percentage scores (D) that included all of the COVID-19 original articles (n = 13) and the lower quality half of the nonCOVID-19 original articles (n = 26). Data are means with 95% CI. An adjusted threshold P value of 0.025 defines significance (adjusted for multiple testing. Welch's t-tests).

In a secondary sensitivity analysis that also included research letters, the mean total scores in the COVID-19 (n = 21) and nonCOVID-19 (n = 55) groups were 12.3 (95% CI 10.6–14) and 23.3 (95% CI 22.2–24.2) respectively, and the mean summary percentage scores were 72.6% (95% CI 66.1–79.1) and 90.9% (95% CI 88.9–92.9), respectively. The mean total score

and the mean summary percentage scores were significantly different between the groups, favoring the nonCOVID-19 original articles (P <0.001, for both; Welch's t-test; Hedges' g = 2.98, 1.87, respectively). For the total scores, the difference between the means was 11.0 (95% CI 9.1–12.9; P <0.001), and for the summary percentage scores, 18.3% (95% CI 11.6%–25.0%; P <0.001).

## Citation frequency

There was a significant difference in the median number of citations according to GoogleScholar at each of the seven dates tested, favoring COVID-19 original research papers (P <0.001, for all; Mann-Whitney, Table 3). A comparison to a set of original articles from the same dates in 2019 revealed 53 (25 to 90) citations in 2019 vs. 334 (222 to 1001) citations for COVID articles in August 2020 and 10 (4 to 18) for non COVID articles (p<0.002 for all comparisons). When corrected for lead-time with citations per month, the articles in 2019 have 4 (2 to 6) cites per month, the non-Covid articles in 2020 2.5 (1 to 4.5) without significance. The COVID articles in 2020 have 83.5 (55 to 250) cites per month (p<0.001).

## Narrative appraisal

The major weaknesses of the 13 COVID-19 original research articles were assessed (Table 4). The selection included one randomized trial [41], four retrospective cohort studies or case series [42–45], five epidemiological descriptive studies [46–50], three epidemiologic modeling studies [51–53], with most of the designs reflecting low grades of evidence [22]. Most of these studies had limitations in terms of missing data or under-reporting. The randomized trial was not blinded. Ten studies showed no apparent conflicts of interest. Two studies were based on data collected by the World Health Organization [51, 52], and in another study [54] a pharmaceutical company screened the patients for treatment, collected the data, and supported the trial financially. Two studies had a patient:author ratio <1 [43, 46]. Two studies were close to 1 [55, 56]. Three studies were considered not relevant for further research [46, 48, 55], and four studies were deemed not relevant for clinical practice [43, 46, 55, 56], because the findings were neither new nor generalizable. The 13 COVID-19 original articles have already been cited in 52 sets of published guidelines.

## Discussion

The main finding of our study is that the COVID-19–related research in these highly ranked medical journals is of lower quality than research on other topics in the same journals for the

**Table 3. Google Scholar citations of original articles published between March 12 and April 12, 2020.**

| Date | Original articles citations | | P value* |
|---|---|---|---|
| | **COVID-19 (n = 13)** | **nonCOVID-19 (n = 52)** | |
| April 25 | 33 (14–212) | 2 (1–3) | <0.001 |
| April 30 | 45 (30–244) | 2 (1–4) | <0.001 |
| May 5 | 65 (41–290) | 2 (1–4) | <0.001 |
| May 10 | 88 (48–328) | 2 (1–5) | <0.001 |
| May 15 | 123 (59–390) | 2.5 (1–5) | <0.001 |
| May 20 | 139 (64–435) | 3 (1.3–6) | <0.001 |
| May 25 | 149 (73–512) | 3 (1.3–7) | <0.001 |

Data are median (interquartile range)

* Mann-Whitney tests

**Table 4. Narrative assessment of the quality of the COVID-19 original articles.**

| Reported study | Major weaknesses | Conflict of interest | Patient: author (ratio) | Should influence further research? | Should influence clinical practice? | Citation rate as of April 30 |
|---|---|---|---|---|---|---|
| **Bhatraju et al**. Covid-19 in critically ill patients in the Seattle region—case series [55] | Design implies a low grade evidence (case-series; no generalizable or representative information). Patients presented with similar respiratory symptoms and had similar mortality rate to patients described in reports from China. Incomplete documentation of symptoms and missing laboratory testing | None apparent | 24:18 (1.33) | No. Similar data across Chinese and European cohorts. | No. No new findings. Incorporated into two guideline documents | 86 |
| **Cao et al**. A trial of lopinavir-ritonavir in adults hospitalized with severe COVID-19 [41] | Some exclusion criteria were vague (physician decision when involved in the trial as not in the best interest of the patients, presence of any condition that would not allow protocol to be followed safely). No blinding. No placebo prepared. | None apparent | 199:65 (3.06) | Yes. Pursuing more trials with lopinavir-ritonavir not necessary. | Yes. Lopinavir-ritonavir treatment added to standard supportive care not associated with clinical improvement or mortality in seriously ill patients with COVID-19, and therefore should not be used for treatment. | 389 |
| **Ghinai et al**. First known person-to-person transmission of severe acute respiratory syndrome coronavirus 2 (SARS-CoV-2) in the USA [46] | Design implies low grade evidence (case-report; no generalizable or representative information). Incomplete documentation. Epidemiological design performed before implementation of CDC guidelines (not comparable to future investigations). | None apparent | 2:38 (0.05) | No. Epidemiological design performed before implementation of CDC guidelines (methodology not comparable to future investigations). | No. Described before in another country. Incorporated into Position Paper on COVID-19 of the EASL-ESCMID | 38 |
| **Gilbert et al**. Preparedness and vulnerability of African countries against importations of COVID-19: a modelling study [51] | Design implies low grade evidence (epidemiologic modeling study; anticipatory). Study did not state limitations. Complex analysis. | Yes. WHO supported | N/A | Yes. Should influence public health measures and research for implementation and effectiveness | Yes. Should influence public health measures. Mainly Africa-derived research | 98 |
| **Grasselli et al**. Baseline characteristics and outcomes of 1591 patients infected with SARS-CoV-2 admitted to ICUs of the Lombardy region, Italy [42] | Design implies low grade evidence (Case-series). Data acquired telephonically. Large amounts of missing data. ICU mortality reported while 58% were still on ICU. | None apparent | 1591:21 (75.76) | Yes. Baseline data for Europe. | Yes. Representative cohort to inform clinical practice. Incorporated into a Position Paper of the German Society of Pneumology on treatment for COVID-19 and in guideline from ENT-UK for safe tracheostomy of COVID-19 patients. | 51 |
| **Grein et al**. Compassionate use of remdesivir for patients with severe COVID-19 [54] | Design implies low grade evidence (Case-Series). No sample size calculation/ small sample size/ underpowered study. Limited number of collected laboratory measures. Missing data. No control group. | Yes. Medication supplied after request to Gilead. Gilead funded trial, collected data, and decided which patients got drug | 53:56 (0.94) | Yes. Findings from these uncontrolled data informed by the ongoing randomized, placebo-controlled trials of remdesivir therapy for COVID-19. | Currently no. Data too low quality to influence clinical practice, concerns regarding patient safety. Included in four sets of guidelines. | 42 |

(*Continued*)

**Table 4.** (Continued)

| Reported study | Major weaknesses | Conflict of interest | Patient: author (ratio) | Should influence further research? | Should influence clinical practice? | Citation rate as of April 30 |
|---|---|---|---|---|---|---|
| **Kandel et al**. Health security capacities in the context of COVID-19 outbreak: an analysis of International Health Regulations annual report data from 182 countries [52] | Design implies a low grade evidence (epidemiologic modelling study; anticipatory). Study does not state limitations. Complex analysis. | Yes. WHO supported | N/A | Yes. Should influence public health measures and research for implementation and effectiveness | Yes. Should influence public health measures and research for implementation and effectiveness. | 24 |
| **Leung et al**. First-wave COVID-19 transmissibility and severity in China outside Hubei after control measures, and second-wave scenario planning: a modelling impact assessment [53] | Design implies a low grade evidence (epidemiologic modelling study; anticipatory). Under-reporting from national sources. Complex analysis. | None apparent | N/A | Yes. Should influence public health measures and research for implementation and effectiveness | Yes. Should influence public health measures and research for implementation and effectiveness. | 11 |
| **Li et al**. Early transmission dynamics in Wuhan, China, of novel Coronavirus-infected pneumonia [47] | Design implies a low grade evidence (epidemiologic descriptive study). Missing values, probably underreporting. | None apparent | 425:45 (9.44) | Yes. First estimate of pandemic dynamics. | Yes. Representative cohort can inform clinical practice. Included in eight sets of guidelines | 2027 |
| **McMichael et al**. Epidemiology of COVID-19 in a long-term care facility in King County, Washington [48] | Design implies a low grade evidence (epidemiologic descriptive study). Missing values. | None apparent | 147/31 (4.74) | No. Similar data to other cohorts, no generalizability of results. | Yes. Representative cohort can inform clinical practice. Included in two societal recommendations for protecting against and mitigation of COVID-19 pandemic in long-term care facilities. | 45 |
| **Pan et al**. Association of public health interventions with the epidemiology of the COVID-19 outbreak in Wuhan, China [49] | Design implies a low grade evidence (epidemiologic descriptive study). Missing values. Questionable findings (letter from Lipsitch et al.) [63] | None apparent | N/A | Yes. Should influence public health measures and research for implementation and effectiveness. | Yes. Should influence public health measures and research for implementation and effectiveness. | 24 |
| **Pung et al**. Investigation of three clusters of COVID-19 in Singapore: implications for surveillance and response measures [50] | Design implies a low grade evidence (epidemiologic descriptive study). Small sample size. Missing values. Recall bias. | None apparent | 36:20 (1.80) | Might influence public health measures to contain clusters. | No. Data too low quality to influence clinical practice (no generalizability). | 36 |
| **Zhou et al**. Clinical course and risk factors for mortality of adult inpatients with COVID-19 in Wuhan, China: a retrospective cohort study [44] | Small sample. Missing values. | None apparent | 191:19 (10.05) | Yes. Early description of clinical course. Findings might change with ongoing pandemic and for other health systems | Yes. Representative cohort can inform clinical practice. Included in 33 sets of guidelines from different societies (all continents represented). | 1085 |

CDC: Center for Disease Control; N/A, not applicable

same period of time, with strong measures for effect size. We also demonstrated that the number of publications on COVID-19 alone is almost the same as the number of publications on all other topics. These findings provide evidence for the debate on the scientific value, ethics, and information overload of COVID-19 research [10, 13, 19].

There are several limitations to the present study. Even though our data were less than a month old at first submission, the results may soon become obsolete, as new COVID-19 research emerges on a daily basis. We tried to overcome potential bias with a clear search strategy and simple analysis, making our findings highly reproducible. We chose Lander's method because it allowed inclusion of *in-vitro* and animal research [23], and we refined the hierarchical grading of the level of evidence using a quantitative tool [27]. Given the vast choice [57], we chose the QUALSYST-tool on the basis that it allows assessment and comparison across multiple study types [27]. Even when the summary scoring might be biased for a methodological quality assessment [57], "composite quality scales can provide useful overall assessments when comparing populations of trials" [57]. The QUALSYST tool has been validated and is easy to use. This may facilitate additional similar studies at a later stage of the pandemic. Compared to an in-depth analysis of a study's peer-review process prior to acceptance for publication, it must remain very superficial. We did not expand our analysis to check source data. The data scandal leading to retraction of two major studies [8, 9] emerged while our article was under peer-review. The tools we used would not be suitable to have detected this. Public data repositories and an "open science" approach may facilitate data validation [58].

The imbalance between the two cohorts in our study might come from a lack of randomized trials and a proliferation of opinion articles and cluster descriptions for the COVID-19 publications. It can be argued that in the early phases of a pandemic, case-defining reports are mandatory for the evolving dynamics of the outbreak and that such studies will suffer from the usual limitations of initial investigations, and will score lower on quality, even when they are carried out to high standards. However, in our secondary analysis, after exclusion of the highest-quality nonCOVID-19 publications, the significant quality difference remained. One might argue that a comparison to a historical control group, for example the same time frame in 2019, when there was no pandemic effect on research, would have been more appropriate. Our hypothesis was that COVID-related research showed lower quality than non-COVID research. A historical control group may introduce a selection bias, since conditions for research then would be clearly different. We would therefore argue that the control group has to be subject to the same conditions as the test group, when methodological quality is assessed. This may be different for other endpoints like total research output. In line with our results, Stefanini et al reported—in an oral presentation at the European Society of Cardiology Congress 2020—similar findings of lower quality associated with COVID-19 in the same journals and timeframe as our work with a historical control group of 2019. So, both historical and contemporary control groups lead to the same conclusions.

The COVID-19 thematic *per se* might have attracted more readers and researchers, which will have led to more citations and greater incorporation into secondary studies, as we have also demonstrated. Such a 'double-whammy' of lower-quality literature and high dissemination potential can have grave consequences, as it might urge clinicians to take actions and use treatments that are compassionately based but supported by little scientific evidence. Indeed, apart from exposing patients to potential side effects of some drugs [46, 59, 60], treatment strategies based on case reports are generally futile [61]. While multiple diagnostic, therapeutic, and preventive interventions for COVID-19 are being trialed [62], clinicians should sometimes resist the wish "to at least do something", and to maintain clinical equipoise while fully gathering and evaluating the data that are available [12, 61]. This responsibility needs to be shared by the high-impact journals, which should continue to maintain publication standards as for other nonCOVID-19 research. It must be acknowledged though, that a citation does not necessarily need to be positive for a study or author, if the context, i. e. criticism or discussions about retractions and corrections, of the citations are considered. This is beyond the scope of our work.

The pandemic took a toll on all aspects of life. Clearly, journal reviewers were restricted in the time they were able to invest into their valuable, voluntary and honorary work. To what extent changes in their practices have occurred is not accessible for us, since the peer-review process was blind and confidential. Assessing of journals with open peer review during the pandemic may shed light on such phenomena, but this was not the scope of our study.

We also demonstrated a worrying trend of increasingly long authorships in lower quality COVID-19 publications, with the almost 'anecdotical' findings of some of the publications actually having more authors than patients [38, 43, 46]. The current demand for publications appears to have led authors to send their COVID-19 findings to higher-impact journals. As the authors of the present report, we are exposed to the same allegations.

At present, we can only issue a plea to both authors and editors to maintain their ethical and moral responsibilities in terms of the International Committee of Medical Journal Editors authorship standards. Being at the forefront of medical discovery, these journals should not publish lower quality findings just to promote citations. The risk of bias and unintended consequences for patients is relevant [61], and scientific standards must not be 'negotiable'[10].

## Conclusions

The quality of the COVID-19–related research in the top three scientific medical journals is below the quality average of these journals. Unfortunately, our numbers do not contribute to a solution as to how to preserve scientific rigor under the pressure of a pandemic.

## Supporting information

**S1 File. Checklist used for the assessment of the quality of the quantitative studies.** Description of data: Detailed criteria are shown for the quality assessment of the quantitative studies.
(DOCX)

**S2 File. Assessor (authors MZ–DB, JBE–BZ) agreements on the qualities of the quantitative studies.** Description of data: Percentage assessor agreement after independent individual scoring and following resolution of disagreements.
(DOCX)

## Acknowledgments

We would like to thank Professor Jukka Takala for revision of the manuscript draft, and Chris Berrie for manuscript editing and help with the language.

## Author Contributions

**Conceptualization:** Marko Zdravkovic, Joana Berger-Estilita, David Berger.

**Data curation:** Marko Zdravkovic, Joana Berger-Estilita, Bogdan Zdravkovic.

**Formal analysis:** Marko Zdravkovic, Joana Berger-Estilita, Bogdan Zdravkovic, David Berger.

**Investigation:** Marko Zdravkovic, Joana Berger-Estilita, Bogdan Zdravkovic.

**Methodology:** Marko Zdravkovic, David Berger.

**Supervision:** David Berger.

**Validation:** Marko Zdravkovic, Joana Berger-Estilita, Bogdan Zdravkovic, David Berger.

**Visualization:** Marko Zdravkovic, Joana Berger-Estilita.

**Writing – original draft:** Marko Zdravkovic, Joana Berger-Estilita, David Berger.

**Writing – review & editing:** Marko Zdravkovic, Joana Berger-Estilita, Bogdan Zdravkovic, David Berger.

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
