## [Decision Letter · Decision Letter 0]

30 Jul 2020

PONE-D-20-14688

Scientific Quality of COVID-19 and SARS CoV-2 Publications in the Highest Impact Medical Journals during the Early Phase of the Pandemic: A Review and Case-Control

PLOS ONE

Dear Dr. Berger,

Thank you for submitting your manuscript to PLOS ONE. After careful consideration, we feel that it has merit but does not fully meet PLOS ONE’s publication criteria as it currently stands. Therefore, we invite you to submit a revised version of the manuscript that addresses the points raised during the review process.

Could  you please pay attention to the following comments made by the reviewers:

Per reviewer 1's suggestion, specify the 3 journals you are looking at and/or mention that these are the top 3 regarding IF within the general medicine segment.Better explain article types of each journal by providing definitions and reasons for in/exclusion. The reader should not need to search on journal websites what each article type entails. You could also include definitions for article types in the legend of Figure 1.For the level of evidence assessment, please do an additional analysis as suggested by Reviewer 1 = just look at the original research papers and see if the level of evidence is different for these article types.Consider adding all research letters for the QUALSYST analysis (not only the JAMA research letters). I agree with the Reviewer 1 that correspondence/research letters contain new findings and are currently oftentimes full manuscripts presented in shorter form. Moreover, additional tables and figures are usually added as supplemental material in NEJM, Lancet and JAMA. Please provide a rationale why this would not be possible when applicable.Make sure to only include papers for which the Levels of Evidence pyramid can be used.  E.g. exclude the JAMA "medical news and perspectives and "piece of my mind" papers. These are not opinion papers.Address using consecutive controls vs historic controls, together with the expected difference.Reviewer 2 suggests a lot of excellent textual revisions, which should be addressed.

We look forward to receiving your revised manuscript.

Kind regards,

Bart Ferket

Academic Editor

PLOS ONE

Journal Requirements:

2. Please provide additional information concerning the qualitative analyses performed. For example, if any rubric, theoretical framework or protocol was followed please describe this in sufficient detail for replication.

3. Please ensure that "systematic review" is incorprated into the title per PLOS ONE submission guidelines.

6.  Thank you for stating the following in the Competing Interests section: 

"Marko Zdravkovic, Bogdan Zdravkovic and Joana Berger-Estilita  have declared that no competing interests exist.

David Berger has read the journal's policy and the authors of this manuscript have the following competing interests:

The Department of Intensive Care Medicine at Inselspital has, or has had in the past, research contracts with Abionic SA, AVA AG, CSEM SA, Cube Dx GmbH, Cyto Sorbents Europe GmbH, Edwards Lifesciences LLC, GE Healthcare, ImaCor Inc., MedImmune LLC, Orion Corporation, Phagenesis Ltd. and research & development/consulting contracts with Edwards Lifesciences LLC, Nestec SA, Wyss Zurich. The money was paid into a departmental fund; Dr Berger received no personal financial gain.

The Department of Intensive Care Medicine has received unrestricted educational grants from the following organizations for organizing a quarterly postgraduate educational symposium, the Berner Forum for Intensive Care (until 2015): Abbott AG, Anandic Medical Systems, Astellas, AstraZeneca, Bard Medica SA, Baxter, B | Braun, CSL Behring, Covidien, Fresenius Kabi, GSK, Lilly, Maquet, MSD, Novartis, Nycomed, Orion Pharma, Pfizer, Pierre Fabre Pharma AG (formerly known as RobaPharm).

The Department of Intensive Care Medicine has received unrestricted educational grants from the following organizations for organizing bi-annual postgraduate courses in the fields of critical care ultrasound, management of ECMO and mechanical ventilation: Abbott AG, Anandic Medical Systems, Bard Medica SA., Bracco, Dräger Schweiz AG, Edwards Lifesciences AG, Fresenius Kabi (Schweiz) AG, Getinge Group Maquet AG, Hamilton Medical AG, Pierre Fabre Pharma AG (formerly known as RobaPharm), PanGas AG Healthcare, Pfizer AG, Orion Pharma, Teleflex Medical GmbH.".

Reviewers' comments:

Reviewer's Responses to Questions

**Comments to the Author**

1. Is the manuscript technically sound, and do the data support the conclusions?

Reviewer #1: Yes

Reviewer #2: Partly

2. Has the statistical analysis been performed appropriately and rigorously? 

Reviewer #1: Yes

Reviewer #2: Yes

3. Have the authors made all data underlying the findings in their manuscript fully available?

Reviewer #1: No

Reviewer #2: Yes

4. Is the manuscript presented in an intelligible fashion and written in standard English?

Reviewer #1: Yes

Reviewer #2: Yes

5. Review Comments to the Author

Reviewer #1: Many thanks for the opportunity to review this paper. Overall, I think this is a good paper and solid analysis that supports the discussion and conclusion provided. I do, however, identify some aspects of the methods and conclusions that require attention or reconsideration.

Was the impact factor >50 prespecified or did that just happen to fit the 3 journals the authors wanted to look at? I can’t imagine that the authors didn’t know exactly what journals would be included by setting the cutoff that high. Since it doesn’t appear this analysis was prespecified just be upfront about the journals you wanted to look at and why.

Page 5 Lines 102-106: This is not presented clearly. For instance the authors say they exclude correspondence but then (rightfully as they may include original results) include some correspondence in their final sample. Please add a bit more detail about what was included and excluded and why.

The Oxford Quality Rating Scheme is a valid choice for the purposes described, but I am wary of one aspect of its use: the inclusion of editorials, which I assume are largely categorised as the lowest level of evidence (“expert opinion”). I fear this might be skewing the findings quite a bit and I’m not sure top journals publishing editorials on the all-consuming topic of the moment, in its early stages when original research is going to be sparse, is indicative of compromising their standards evidence (the authors allude to this in their discussion). I would be interested in a sensitivity analysis that excluded these and seeing how it might impact the robustness of the findings.

The authors exclude JAMA Research Letters from the “Original Article” count but I think this is a mistake. These are usually original research simply presented in a shorter format for manuscripts that don’t require a full article of detail. However, the level of detail they usually describe should be enough to perform the assessments. If they didn’t, that itself might be an interesting finding on the format (although I don’t think that is the case).

Medical news and perspectives from JAMA are included in the quality assessments. This is odd to me since, as news pieces, they do not really fit as something appropriate to be examined by a “levels of evidence” tool. News is typically not held to the same standard as original research.

The QUALSYST checklist appears to be an acceptable and general enough tool for the purposes described.

Page 6 Lines 140-142: Small detail but were disagreements resolved through consensus with the full group or between each 2-author pair?

Why, for the “Qualitative analysis of COVID-19 original articles” did the authors rely on a personal subjective interpretation of article quality rather than existing methods, like say the Cochrane risk of bias tools (even if modified or slimmed down), to assess article quality? These cover many of the same areas examined and have been created for a wide array of research types.

The citation frequency analysis is the part of the study that I feel the least confident drawing any meaningful conclusions from as a reader. COVID articles are extremely prevalent and concentrated in a single area of great interest while the rest of the corpus is likely spread out over the entire rest of the field of biomedical sciences. With a preponderance of research being published in these journals dealing with COVID specifically, a high citation rate feels natural rather than indicative of any larger issue. Similarly, more than usual, I think the speed at which the academic community is presenting hypothesis and exploratory research on an entirely novel disease, and focusing intense public scrutiny on notable findings, may be leading to a situation in which the context around a citation is crucial to understanding this research question. I could reasonably imagine that a given study or article is far more likely to be cited alongside criticism, or in some negative light, than usual in the context of COVID research. This might be something the authors could explore given the relatively small amount of citations (52) included in their analysis.

Please make sure you discuss all your reported outcomes in the methodology section. For instance, you report the amount of authors without ever mentioning this was being examined in the methods section.

Would Hedge’s g be a more accurate estimator of effect size in this case than Cohen’s d given the unequal sample sizes? Though since the “sample” is complete for articles from these journals for this time period, perhaps Cohen’s d is appropriate? I’m not entirely sure one way or the other but it might be worth the authors justifying their use of one vs the other so it is clear to the reader.

The authors should report the actual mean total and summary percentage scores (and resulting error bars) and not just refer to Figure 2 from which it is difficult to tell the exact numbers.

It is good that the authors are willing to make their data available, however I don’t see any reason the underlying data for this study couldn’t be proactively made available on an appropriate repository like OSF or Figshare (or any other appropriate public repository) rather than from the author. I would think this would be particularly important for this research as others may want to pick up and expand on these methods as the pandemic matures. Requiring interested parties to contact the author is simply an additional barrier to the benefits of data sharing and availability. I would ask the authors to consider taking this proactive step.

The authors recognize what I believe to be the biggest issue with this paper which is timeliness. Obviously there is not much to be done to combat this other than a potential update of the results before submission but this would be a lot to ask. However, I do agree this is a major limitation of these findings in terms of overall interest to the community. That said, I think even more value can come from sharing the framework for the methods to potentially be expanded and applied at other points in the epidemic. I think there is some interesting potential for this that the authors would do well to acknowledge.

The authors appropriately acknowledge the inherent limitations of choosing any single assessment tool for paper quality and I think their choice to use QUALSYST is fine. However the recent Surgisphere scandal brings this limitation to further light. I’m fairly certain this methodology would have assessed both the NEJM and Lancet Surisphere papers that were retracted as high quality research despite failing on some general quality measures. Things like the availability of data and materials are metrics that aren’t included in the scales used that may also be important indicators of quality. To be clear, I do not think it would be reasonable for the authors to have scrutinized the papers included to a level that they would have noticed the irregularities that tipped off the community to the issues with the Surgisphere Lancet paper. That level of detailed scrutiny was not within the scope of this paper. However there are broad indicators largely related to best practice in Open Science (and I’m sure in some other areas as well) that impact their ability to assess quality. Just an interesting case study to consider in grounding this limitation.

Reviewer #2: I have provided a review of the manuscript "Scientific Quality of COVID-19 and SARS CoV-2 Publications in the Highest Impact Medical Journals during the Early Phase of the Pandemic: A Review and Case-Control”

The authors provide am interesting manuscript which can enlighten the quality of COVID-19 related publications which could have been affected by rapidity and high needs in this pandemic.

INTRODUCTION

Line 65 – 68. Can be worthy to add the example of the retraction of the papers published on The Lancet and New England Journal regarding the use of Covid-19 treatments chloroquine and hydroxychloroquine https://www.thelancet.com/journals/lancet/article/PIIS0140-6736(20)31180-6/fulltext. This info can strengthen the rational of the study.

Lines 86-87: “We hypothesized that the quality of recent publications on COVID-19 in medical journals with impact factor >50 is lower than for nonCOVID-19 articles published during the same time period.”

I wonder if considering the same period as span of time for the non-covid publications is adequate. COVID19 has affected not only the publications related to it but also all the other specialties have been altered and affected. Priorities has changed during the first 6 months of 2020 and also the publications process has been speeded for COVID19 publications but slowed for the non-COVID19 areas. Would have been more appropriate to consider the same months for the non-covid publications but in 2019 when COVID19 couldn’t affected the quality and the process?

METHODS

Lines 91 – 92. “This report follows the applicable STROBE guidelines for case-control studies and PRISMA guidelines for systematic reviews”. Could you please define the study design? Is this an observational study (ie. Case-control or even cross sectional) or a review? Why use the PRISMA guideline if not properly a systematic review? This is not a systematic review but a case/control study so observational.

Lines 102 – 104. “The resulting publications were stratified into COVID-19–related and nonCOVID-19–related. We matched the nonCOVID-19 publications with COVID-19 publications according to article types within each journal…”.

As already discussed above, wouldn’t an historical control, ie non-COVID19 publication from March 12 to April 12, 2019 be more appropriate and unbiased? The publications process for the non-covid19 has been altered due to the COVID19 priprities. For example, peer review process were slower for non-covid19 than covid-19 publications; the attention were more directed to COVID19 publications than non COVID.

Table 1. Table 1 should be included only in results sections not in the methods since it reported characteristics as level of evidence of the included studies.

Lines 126-127. “Quantitative appraisal” and lines “Qualitative analysis”. Reading these two sections, I would not find a difference in terms of quantitative or qualitative appraisal. The checklist the authors used to evaluate the methodological quality express a qualitative approach and not a quantitative one. Quantitative appraisal or quantitative synthesis usually refers to a meta-analysis or any intended statistical methods for combining results across studies (e.g. meta-analysis, subgroup analysis, meta-regression, sensitivity analysis), including methods for assessing heterogeneity

In this case, it may be more informative to write about “qualitative appraisal of quantitative research” or simply report a unique paragraph with the qualitative appraisal. Then, what reported in lines 144 – 147 (ie. Funding or missing data) can be an other way to inform qualitatively about the included research.

Line 174: “Quantitative appraisal of the quality of the original articles is..” I would change this terms into qualitative assessment of original articles. Quality is implicitly considered into “qualitative assessment”.

Line 232: “favoring COVID-19 original research papers”. I suppose that this might be obvious considering the period of high demand for COVID19 answers in the international scientific community. A comparison with non-COVID19 research in a period not affected by COVID19 could have been more appropriate for detecting the real difference in number of citations.

Lines 239-240: “Most of these studies had limitations in terms of missing data or under-reporting. The randomized trial was not blinded” This sentence confused me. What did the authors consider as quantitative and what as qualitative assessment? Elements considered in the “qualitative assessment” are related to the methodological quality of the study designs. For example, the blinding in a RCT is an item of the Cochrane Risk of Bias tool whose aim is to assess the internal validity of a randomized controlled trial, the risk of bias in terms of methodological quality. https://handbook-5-1.cochrane.org/chapter_8/8_assessing_risk_of_bias_in_included_studies.htm

DISCUSSION

Discussion is too limited and need to be enriched and enlarged – several issues can be of interest, here some already arisen in the issues above:

- The assessment of nonCOVID19 publication in a different period could have changed the results?

- The period is highly influenced by the changed research priorities related to COVID19 – the efforts (money, time etc…) of the whole international scientific community has been dedicated to COVID19.

- Could the journal peer review process have affected the quality of the published journal? In this six months of SARS-COV-2 pandemic even the attention of editors and reviewer was directed to speed as much as possible COVID19 publications: did the author discuss this? Moreover, a lot of pre-prints on COVID19 exist – could these influence the publications of COVID-19 research?

6. PLOS authors have the option to publish the peer review history of their article (what does this mean?). If published, this will include your full peer review and any attached files.

Reviewer #1: **Yes: **Nicholas J. DeVito

Reviewer #2: No

---

## [Author Response · Author response to Decision Letter 0]

21 Aug 2020

Reply to reviewers for PONE-D-20-14688-R1: 

Scientific Quality of COVID-19 and SARS CoV-2 Publications in the Highest Impact Medical Journals during the Early Phase of the Pandemic: A Review and Case-Control study

Editor’s comments

Thank you for submitting your manuscript to PLOS ONE. After careful consideration, we feel that it has merit but does not fully meet PLOS ONE’s publication criteria as it currently stands. Therefore, we invite you to submit a revised version of the manuscript that addresses the points raised during the review process.

Could you please pay attention to the following comments made by the reviewers:

1. Per reviewer 1's suggestion, specify the 3 journals you are looking at and/or mention that these are the top 3 regarding IF within the general medicine segment.

Please refer to our answer A-1-1

2. Better explain article types of each journal by providing definitions and reasons for in/exclusion. The reader should not need to search on journal websites what each article type entails. You could also include definitions for article types in the legend of Figure 1.

Please refer to our answer A-1-2

3. For the level of evidence assessment, please do an additional analysis as suggested by Reviewer 1 = just look at the original research papers and see if the level of evidence is different for these article types.

Please refer to our Answer A-1-3. 

4. Consider adding all research letters for the QUALSYST analysis (not only the JAMA research letters). I agree with the Reviewer 1 that correspondence/research letters contain new findings and are currently oftentimes full manuscripts presented in shorter form. Moreover, additional tables and figures are usually added as supplemental material in NEJM, Lancet and JAMA. Please provide a rationale why this would not be possible when applicable.

Please refer to our Answer A-1-4. 

5. Make sure to only include papers for which the Levels of Evidence pyramid can be used. E.g. exclude the JAMA "medical news and perspectives and "piece of my mind" papers. These are not opinion papers.

Please refer to our Answer A-1-5. 

6. Address using consecutive controls vs historic controls, together with the expected difference.

Please refer to our Answer A-2-2. We think that selecting a different time period for assessment of quality would introduce a selection bias, since conditions are not the same. We have added a historic citation count.

7. Reviewer 2 suggests a lot of excellent textual revisions, which should be addressed.

Please refer to our last anwer in the reply letter and to the expanced discussion.

Reviewer #1: 

Many thanks for the opportunity to review this paper. Overall, I think this is a good paper and solid analysis that supports the discussion and conclusion provided. I do, however, identify some aspects of the methods and conclusions that require attention or reconsideration.

We thank the reviewer for the appreciation of our work

Q- 1-1: Was the impact factor >50 prespecified or did that just happen to fit the 3 journals the authors wanted to look at? I can’t imagine that the authors didn’t know exactly what journals would be included by setting the cutoff that high. Since it doesn’t appear this analysis was prespecified just be upfront about the journals you wanted to look at and why.

A-1-1: We choose the three journals as you rightfully say based on their reputation and influence, rather than on a prespecified impact factor.

The Introduction and Methods were adapted as follows:

“To date, the quality of COVID-19 publications in the top three general medical journals by impact factor (i. e. the New England Journal of Medicine, The Lancet and The Journal of the American Medical Association, represented by an impact factor > 50 for all) has not been formally assessed. We hypothesized that the quality of recent publications on COVID-19 in the three most influential medical journals is lower than for nonCOVID-19 articles published during the same time period.”

“For the time period of March 12 to April 12, 2020 (i.e., during the early outbreak phase of the COVID-19 pandemic), we identified all of the publications from the top three general medical journals by impact factor (the New England Journal of Medicine (NEJM), the Journal of the American Medical Association (JAMA), and The Lancet).”

Q-1-2: Page 5 Lines 102-106: This is not presented clearly. For instance the authors say they exclude correspondence but then (rightfully as they may include original results) include some correspondence in their final sample. Please add a bit more detail about what was included and excluded and why.

A1-2: We have excluded only correspondence letters on previously published papers – which we believe is now clear from text and Figure 1. 

“Secondary studies, correspondence letters on previously published articles, unauthored publications, and specific article types not matching any of the six categories on the levels of the evidence pyramid(1-3) (e.g., infographic, erratum) were excluded (Figure 1).” 

Q-1-3: The Oxford Quality Rating Scheme is a valid choice for the purposes described, but I am wary of one aspect of its use: the inclusion of editorials, which I assume are largely categorised as the lowest level of evidence (“expert opinion”). I fear this might be skewing the findings quite a bit and I’m not sure top journals publishing editorials on the all-consuming topic of the moment, in its early stages when original research is going to be sparse, is indicative of compromising their standards evidence (the authors allude to this in their discussion). I would be interested in a sensitivity analysis that excluded these and seeing how it might impact the robustness of the findings.

A-1-3: This is a valid argument, but we would like to draw your attention to the numbers: there are 12 editorials in the COVID group included in the analysis (out of 155 = 7.7%), and there are 18 editorials in the nonCOVID group (out of 130 = 13.8%). Given these numbers, we think that the editorials are not likely to skew the data in the direction mentioned above. Also, we wanted to compare all the levels of evidence represented – including the opinion papers in the initial stage as evident from Table 1 and as explained in the methods section:

"We matched the nonCOVID-19 publications with COVID-19 publications according to article types within each journal, with the exclusion of nonmatching article types."

Then in the second stage we analysed, in greater detail, the original research papers only, which are supposed to be of the highest quality and again showed major differences between the two groups of papers. 

We have now added a sensitivity analysis upon your request in table 2 which only included full original articles. It clearly confirms our hypothesis. The results section was adapted accordingly.

“When comparing only the original articles on the levels of evidence pyramid (Table 2), the nonCOVID-19 publications were also associated with higher quality (P <0.001; Chi squared), with a strong association measure (Cramer's V: 0.641, Table 2). When comparing the higher evidence group to the lower evidence group, the COVID-19 original articles were 26-fold more likely (i.e., odds ratio) to be in the lower evidence group (95% CI: 5.4–120; P <0.001).”

Q-1-4: The authors exclude JAMA Research Letters from the “Original Article” count but I think this is a mistake. These are usually original research simply presented in a shorter format for manuscripts that don’t require a full article of detail. However, the level of detail they usually describe should be enough to perform the assessments. If they didn’t, that itself might be an interesting finding on the format (although I don’t think that is the case).

A-1-4: This is an interesting point raised here. As we discussed this within the group in the planning phase and decided not to include them in Original papers group because: 1) These were decided by the editors/reviewers as not to be good enough to meet the criteria for Original article category (a separate article type in JAMA); 2) There were 8 COVID research letters and 3 nonCOVID published in JAMA – would clearly skew the quality of original papers 3); JAMA is the only journal which has this category of papers, but all three journals have a category of “Original research”. 

We have now provided a separate sensitivity analysis for original papers below *including 11 JAMA research letters* in the main text of results.

“The mean total scores in the COVID-19 (n=21) and nonCOVID-19 (n=55) groups were 12.3 (95% CI 10.6-14) and 23.3 (95% CI 22.2-24.2) respectively, and the mean summary percentage scores were 72.6% (95% CI 66.1-79.1) and 90.9% (95% CI 88.9-92.9), respectively. The mean total score and the mean summary percentage scores were significantly different between the groups, favoring the nonCOVID-19 original articles (P <0.001, for both; Welch's t-test; Hedges’ g = 2.98, 1.87, respectively). For the total scores, the difference between the means was 11.0 (95% CI 9.1–12.9; P <0.001), and for the summary percentage scores, 18.3% (95% CI 11.6%–25.0%; P <0.001). “

Q-1-5: Medical news and perspectives from JAMA are included in the quality assessments. This is odd to me since, as news pieces, they do not really fit as something appropriate to be examined by a “levels of evidence” tool. News is typically not held to the same standard as original research.

A-1-5: We followed the suggestion and adapted the data accordingly to the reviewers comment. Figure 1 and the appropriate positions in the text have been edited accordingly.

Q-1-6: The QUALSYST checklist appears to be an acceptable and general enough tool for the purposes described.

Page 6 Lines 140-142: Small detail but were disagreements resolved through consensus with the full group or between each 2-author pair?

A-1-6a: This has now been clarified:

“Disagreements between the reviewers (defined as >2 difference in the total score, or >10% difference in the summary percentage scores), were resolved through one round of discussion between each 2-author pair. “

Why, for the “Qualitative analysis of COVID-19 original articles” did the authors rely on a personal subjective interpretation of article quality rather than existing methods, like say the Cochrane risk of bias tools (even if modified or slimmed down), to assess article quality? These cover many of the same areas examined and have been created for a wide array of research types.

A-1-6b: Thank you for this insightful comment. In fact, there seems to be a confusion between qualitative analysis as a methodology and analysis of the quality of the papers. We used the QUALSYST checklist as a tool to characterize the quality of the papers. This tool is similar (in terms of its function) to the Cochrane Risk of Bias tool. The reasons for choosing the QUALSYST have been described in the paper.

Qualitative Methodology, on the other hand, uses subjective judgment to analyze a value based on non-quantifiable information. Qualitative analysis is the analysis of qualitative data such as text data from interview transcripts. Unlike quantitative analysis, which is statistics-driven and largely independent of the researcher, qualitative analysis is heavily dependent on the researcher’s analytic and integrative skills and personal knowledge of the social context where the data is collected. The emphasis in qualitative analysis is “sense making” or understanding a phenomenon, rather than predicting or explaining. This methodology is very frequently used in social sciences in combination with quantitative analysis, the so called mixed-methods methodology. We decided to perform qualitative analysis in our work to allow for data triangulation, therefore strengthening our hypothesis(4). We have adapted the method section to clarifiy these differences. 

 “Mixed methods design 

We performed a multi-step 360-degree assessment of the studies. It consisted of their classification according to level of evidence for a quantitative appraisal of their methodological quality using a validated tool, and a qualitative analysis of the strengths and weaknesses of the COVID-19 publications, as is often used in social sciences(4) Early citation frequencies of the original articles was determined. “

Please also refer to reviewer comment Q-2-5 and Q-2-6.

Q-1-7: The citation frequency analysis is the part of the study that I feel the least confident drawing any meaningful conclusions from as a reader. COVID articles are extremely prevalent and concentrated in a single area of great interest while the rest of the corpus is likely spread out over the entire rest of the field of biomedical sciences. With a preponderance of research being published in these journals dealing with COVID specifically, a high citation rate feels natural rather than indicative of any larger issue. Similarly, more than usual, I think the speed at which the academic community is presenting hypothesis and exploratory research on an entirely novel disease, and focusing intense public scrutiny on notable findings, may be leading to a situation in which the context around a citation is crucial to understanding this research question. I could reasonably imagine that a given study or article is far more likely to be cited alongside criticism, or in some negative light, than usual in the context of COVID research. This might be something the authors could explore given the relatively small amount of citations (52) included in their analysis.

A-1-7: Thank you for this comment. We have followed this up for a month and provided a new table here. Your suggestion for checking the content of the citing papers is great, the problem is that the number of total citations is not small – median for COVID papers was 149 on May 25th for the 13 papers / the total number of citations to check on May 25th would be 7,500. This would be an interesting study though on itself given the amount of text to be screened for the content of citations, especially on COVID publications. Please respect that this is beyond the capacity of our small group.

Q-1-8: Please make sure you discuss all your reported outcomes in the methodology section. For instance, you report the amount of authors without ever mentioning this was being examined in the methods section.

A-1-8: We thank you for making us aware of this omission

The method section has been adapted accordingly

“The number of authors per publication was counted manually.”

Q-1-9: Would Hedge’s g be a more accurate estimator of effect size in this case than Cohen’s d given the unequal sample sizes? Though since the “sample” is complete for articles from these journals for this time period, perhaps Cohen’s d is appropriate? I’m not entirely sure one way or the other but it might be worth the authors justifying their use of one vs the other so it is clear to the reader.

A-1-9: We thank you for this suggestion as Hedges g is indeed more robust for small sample sizes. The statistical section and results have been updated accordingly.

“Hedges’s g is used as the effect size measure based on a standardized mean difference (5)(small: d = 0.20; medium: d = 0.50; large: d = 0.80; very large: d = 1.20; huge: d >2.00)(6)»

Q-1-10: The authors should report the actual mean total and summary percentage scores (and resulting error bars) and not just refer to Figure 2 from which it is difficult to tell the exact numbers.

A-1-10: We have provided the requested numbers in the results section.

Q-1-11: It is good that the authors are willing to make their data available, however I don’t see any reason the underlying data for this study couldn’t be proactively made available on an appropriate repository like OSF or Figshare (or any other appropriate public repository) rather than from the author. I would think this would be particularly important for this research as others may want to pick up and expand on these methods as the pandemic matures. Requiring interested parties to contact the author is simply an additional barrier to the benefits of data sharing and availability. I would ask the authors to consider taking this proactive step.

A-1-11: Thank you for this comment. We have decided to make all our data available on Figshare. https://figshare.com/projects/Scientific_Quality_of_COVID-19_and_SARS_CoV-2_Publications_in_the_Highest_Impact_Medical_Journals_during_the_Early_Phase_of_the_Pandemic_A_Case-Control_Study/86027

Q-1-12: The authors recognize what I believe to be the biggest issue with this paper which is timeliness. Obviously there is not much to be done to combat this other than a potential update of the results before submission but this would be a lot to ask. However, I do agree this is a major limitation of these findings in terms of overall interest to the community. That said, I think even more value can come from sharing the framework for the methods to potentially be expanded and applied at other points in the epidemic. I think there is some interesting potential for this that the authors would do well to acknowledge.

A-1-12: We thank you for recognizing that updating our article with additional five months of pandemic publications would simply be beyond our capabilities. The issue of timeliness was demonstrated to us clearly by the emerging Surgisphere scandal, during which our article was “hidden” in peer review. We have expanded the discussion in this direction, please also refer to our next answer.

Q-1-13: The authors appropriately acknowledge the inherent limitations of choosing any single assessment tool for paper quality and I think their choice to use QUALSYST is fine. However the recent Surgisphere scandal brings this limitation to further light. I’m fairly certain this methodology would have assessed both the NEJM and Lancet Surisphere papers that were retracted as high quality research despite failing on some general quality measures. Things like the availability of data and materials are metrics that aren’t included in the scales used that may also be important indicators of quality. To be clear, I do not think it would be reasonable for the authors to have scrutinized the papers included to a level that they would have noticed the irregularities that tipped off the community to the issues with the Surgisphere Lancet paper. That level of detailed scrutiny was not within the scope of this paper. However there are broad indicators largely related to best practice in Open Science (and I’m sure in some other areas as well) that impact their ability to assess quality. Just an interesting case study to consider in grounding this limitation.

A-1-13: We fully agree. The Surgisphere scandal emerged while our article was under review. Please also refer to the first comment fo reviewer 2. 

This underscores the issue of timeliness from above. We have expanded the discussion on timeliness, discuss our methodology in the light of source data and open science.

“The QUALSYST tool has been validated and is easy to use. This may facilitate additional similar studies at a later stage of the pandemic. Compared to an in-depth analysis of a study’s peer-review process prior to acceptance for publication, it must remain very superficial. We did not expand our analysis to check source data. The data scandal leading to retraction of two major studies (7, 8) emerged while our article was under peer-review. The tools we used would not be suitable to have detected this. Public data repositories and an “open science” approach may facilitate data validation(9).”

“The pandemic took a toll on all aspects of life. Clearly, journal reviewers were restricted in the time they were able to invest into their valuable, voluntary and honorary work. To what extent changes in their practices have occurred is not accessible for us, since the peer-review process was blind and confidential. Assessing of journals with open peer review during the pandemic may shed light on such phenomena, but this was not the scope of our study.” 

Reviewer #2: 

I have provided a review of the manuscript "Scientific Quality of COVID-19 and SARS CoV-2 Publications in the Highest Impact Medical Journals during the Early Phase of the Pandemic: A Review and Case-Control”

The authors provide am interesting manuscript which can enlighten the quality of COVID-19 related publications which could have been affected by rapidity and high needs in this pandemic.

We thank the referee for the appreciation of our work.

INTRODUCTION

Q-2-1: Line 65 – 68. Can be worthy to add the example of the retraction of the papers published on The Lancet and New England Journal regarding the use of Covid-19 treatments chloroquine and hydroxychloroquine https://www.thelancet.com/journals/lancet/article/PIIS0140-6736(20)31180-6/fulltext. This info can strengthen the rational of the study.

A-2-1: We thank the reviewer for the opportunity to add a bit of detail on the Surgisphere scandal, which emerged while our article was under review. We have adapted the introduction according to your suggestion. Please also refer to answer 13 for reviewer 1.

“The first report of COVID-19 transmission in asymptomatic individuals(10) was later considered to have been flawed, because the patient showed symptoms at the time of transmission(11). A similar example occurred in The Lancet, whereby the authors retracted a publication after admitting irregularities on the first-hand account of the front-line experience of two Chinese nurses(12). While our article was under review, two major analyses on the use of hydroxychloroquine and cardiovascular mortality associated with COVID-19 were retracted in the Lancet(8) and the New England Journal of Medicine(7) because source data could not be verified. “

Q-2-2: Lines 86-87: “We hypothesized that the quality of recent publications on COVID-19 in medical journals with impact factor >50 is lower than for nonCOVID-19 articles published during the same time period.”

I wonder if considering the same period as span of time for the non-covid publications is adequate. COVID19 has affected not only the publications related to it but also all the other specialties have been altered and affected. Priorities has changed during the first 6 months of 2020 and also the publications process has been speeded for COVID19 publications but slowed for the non-COVID19 areas. Would have been more appropriate to consider the same months for the non-covid publications but in 2019 when COVID19 couldn’t affected the quality and the process?

A-2-2: We clearly see your point and agree that the pandemic has affected all areas of research. The redirection of efforts to COVD-19 may have lowered the output of non-Covid research and slowed down the publication and review process. This was not our research question.

Since we had put the weight in our main analysis on the methodological quality of articles, we think it is important to perform a comparison to non-COVID articles in the same time frame. Our hypothesis was that COVID-related research shows lower quality than non-COVID research. If we had chosen a historical control time, one would introduce a selection bias in the historical control, since conditions for research then were clearly different. We find it important that the control group is subject to the same conditions as the test group and hope that you can follow our reasoning. This issue is now discussed.For other endpoints than quality, like for example total research output in an area or citations, a historical control may be more appropriate. For the citations, we have added data.

“A comparison to a set of original articles from the same dates in 2019 revealed 53 (25 to 90) citations in 2019 vs. 334 (222 to 1001) citations for COVID articles in August 2020 and 10 (4 to 18) for non COVID articles (p<0.002 for all comparisons). When corrected for lead time with citations per month, the articles in 2019 have 4 (2 to 6) cites per month, the non-Covid articles in 2020 2.5 (1 to 4.5) without significance. The COVID articles in 2020 have 83.5 (55 to 250) cites per month (p<0.001).”

“One migh argue that a comparison to a historical control group, for example the same time frame in 2019, when there was no pandemic effect on research, would have been more appropriate. Our hypothesis was that COVID-related research showed lower quality than non-COVID research. A historical control group may introduce a selection bias, since conditions for research then would be clearly different. The control group has to be subject to the same conditions as the test group, when methodological quality is assessed. This may be different for other endpoints like total research output. 

METHODS

Q-2-3: Lines 91 – 92. “This report follows the applicable STROBE guidelines for case-control studies and PRISMA guidelines for systematic reviews”. Could you please define the study design? Is this an observational study (ie. Case-control or even cross sectional) or a review? Why use the PRISMA guideline if not properly a systematic review? This is not a systematic review but a case/control study so observational.

A-2-3: We fully agree with the reviewer. We had planed an observational study design as a case control. We were then obliged by PLOS ONE to include “systematic review” in the title, because we deal with study comparisons. We would like to follow your suggestion to go as a case-control study which was the initial plan, unless the editorial office overrules us.

Q-2-4: Lines 102 – 104. “The resulting publications were stratified into COVID-19–related and nonCOVID-19–related. We matched the nonCOVID-19 publications with COVID-19 publications according to article types within each journal…”.

As already discussed above, wouldn’t an historical control, ie non-COVID19 publication from March 12 to April 12, 2019 be more appropriate and unbiased? The publications process for the non-covid19 has been altered due to the COVID19 priprities. For example, peer review process were slower for non-covid19 than covid-19 publications; the attention were more directed to COVID19 publications than non COVID.

A-2-4: Please refer to our answer to your comment Q-2-2

Q-2-4: Table 1. Table 1 should be included only in results sections not in the methods since it reported characteristics as level of evidence of the included studies.

A-2-4: Thank you. This has been adapted.

Q-2-5: Lines 126-127. “Quantitative appraisal” and lines “Qualitative analysis”. Reading these two sections, I would not find a difference in terms of quantitative or qualitative appraisal. The checklist the authors used to evaluate the methodological quality express a qualitative approach and not a quantitative one. Quantitative appraisal or quantitative synthesis usually refers to a meta-analysis or any intended statistical methods for combining results across studies (e.g. meta-analysis, subgroup analysis, meta-regression, sensitivity analysis), including methods for assessing heterogeneity

In this case, it may be more informative to write about “qualitative appraisal of quantitative research” or simply report a unique paragraph with the qualitative appraisal. Then, what reported in lines 144 – 147 (ie. Funding or missing data) can be an other way to inform qualitatively about the included research.

A-2-5: Thank you for this insightful comment. In fact, there seems to be a confusion between qualitative analysis as a methodology and analysis of the quality of the papers. Please also refer to question 1-6b of reviewer 1 and the adaption in the method section.

Q-2-6: Line 174: “Quantitative appraisal of the quality of the original articles is..” I would change this terms into qualitative assessment of original articles. Quality is implicitly considered into “qualitative assessment”.

A-2-6: We have inserted the suggested changes. Please also refer to answer A1-6b and 2-5.

“Assessment of the original articles’ quality is reported as a two-reviewer mean score (95% CI) and was analyzed using Welch’s t-tests.”

Q-2-7: “favoring COVID-19 original research papers”. I suppose that this might be obvious considering the period of high demand for COVID19 answers in the international scientific community. A comparison with non-COVID19 research in a period not affected by COVID19 could have been more appropriate for detecting the real difference in number of citations.

A-2-7: Please refer to our answer to your comment Q-2-2

Q-2-8: “Most of these studies had limitations in terms of missing data or under-reporting. The randomized trial was not blinded” This sentence confused me. What did the authors consider as quantitative and what as qualitative assessment? Elements considered in the “qualitative assessment” are related to the methodological quality of the study designs. For example, the blinding in a RCT is an item of the Cochrane Risk of Bias tool whose aim is to assess the internal validity of a randomized controlled trial, the risk of bias in terms of methodological quality. https://handbook-5-1.cochrane.org/chapter_8/8_assessing_risk_of_bias_in_included_studies.htm

A-2-8: Please refer to our answer for A1-6b for reviewer 1

DISCUSSION

Q-2-9: Discussion is too limited and need to be enriched and enlarged – several issues can be of interest, here some already arisen in the issues above:

- The assessment of nonCOVID19 publication in a different period could have changed the results?

Please see our response to your question Q 2-2

- The period is highly influenced by the changed research priorities related to COVID19 – the efforts (money, time etc…) of the whole international scientific community has been dedicated to COVID19.

- Could the journal peer review process have affected the quality of the published journal? In this six months of SARS-COV-2 pandemic even the attention of editors and reviewer was directed to speed as much as possible COVID19 publications: did the author discuss this? Moreover, a lot of pre-prints on COVID19 exist – could these influence the publications of COVID-19 research?

This is now discussed

“The pandemic took a toll on all aspects of life. Clearly, journal reviewers were restricted in the time they were able to invest into their valuable, voluntary and honorary work. To what extent changes in their practices have occurred is not accessible for us, since the peer-review process was blind and confidential. Assessing of journals with open peer review during the pandemic may shed light on such phenomena, but this was not the scope of our study.”

1. Phillips B, Ball C, Sackett D, Badenoch D, Straus S, Haynes B, et al. Oxford centre for evidence-based medicine-levels of evidence (March 2009). 2009.

2. Lander B, Balka E. Exploring How Evidence is Used in Care Through an Organizational Ethnography of Two Teaching Hospitals. Journal of medical Internet research. 2019;21(3):e10769.

3. Djulbegovic B, Guyatt GH. Progress in evidence-based medicine: a quarter century on. Lancet. 2017;390(10092):415-23.

4. Understanding qualitative research in health care. Drug and Therapeutics Bulletin. 2017;55(2):21.

5. Durlak JA. How to Select, Calculate, and Interpret Effect Sizes. Journal of Pediatric Psychology. 2009;34(9):917-28.

6. Sawilowsky SS. New effect size rules of thumb. J Mod Appl Stat Methods. 2009;8(2):26.

7. Mehra MR, Desai SS, Kuy S, Henry TD, Patel AN. Retraction: Cardiovascular Disease, Drug Therapy, and Mortality in Covid-19. N Engl J Med. DOI: 10.1056/NEJMoa2007621. New England Journal of Medicine. 2020;382(26):2582-.

8. Mehra MR, Ruschitzka F, Patel AN. Retraction—Hydroxychloroquine or chloroquine with or without a macrolide for treatment of COVID-19: a multinational registry analysis. The Lancet. 2020;395(10240):1820.

9. Shamoo AE. Validate the integrity of research data on COVID 19. Accountability in Research. 2020;27(6):325-6.

10. Rothe C, Schunk M, Sothmann P, Bretzel G, Froeschl G, Wallrauch C, et al. Transmission of 2019-nCoV Infection from an Asymptomatic Contact in Germany. N Eng J Med. 2020;382(10):970-1.

11. Kupferschmidt K. Study claiming new coronavirus can be transmitted by people without symptoms was flawed sciencemag.org2020 [19th April, 2020]. Available from: https://www.sciencemag.org/news/2020/02/paper-non-symptomatic-patient-transmitting-coronavirus-wrong.

12. Zeng Y, Zhen Y. RETRACTED: Chinese medical staff request international medical assistance in fighting against COVID-19. Lancet Glob Health. 2020.

---

## [Decision Letter · Decision Letter 1]

14 Oct 2020

PONE-D-20-14688R1

Scientific Quality of COVID-19 and SARS CoV-2 Publications in the Highest Impact Medical Journals during the Early Phase of the Pandemic: A Case Control Study

PLOS ONE

Dear Dr. Berger,

Thank you for submitting your manuscript to PLOS ONE. After careful consideration, we feel that it has merit but does not fully meet PLOS ONE’s publication criteria as it currently stands. Therefore, we invite you to submit a revised version of the manuscript that addresses the points raised during the review process.

I agree with reviewer 2 regarding the appropriateness of the current title. I also do agree with changing the phrasing of the qualitative appraisal sections. Please use different terminology to describe this analysis, for example narrative appraisal. 

We look forward to receiving your revised manuscript.

Kind regards,

Bart Ferket

Academic Editor

PLOS ONE

Reviewers' comments:

Reviewer's Responses to Questions

**Comments to the Author**

1. If the authors have adequately addressed your comments raised in a previous round of review and you feel that this manuscript is now acceptable for publication, you may indicate that here to bypass the “Comments to the Author” section, enter your conflict of interest statement in the “Confidential to Editor” section, and submit your "Accept" recommendation.

Reviewer #1: (No Response)

Reviewer #2: (No Response)

2. Is the manuscript technically sound, and do the data support the conclusions?

Reviewer #1: Yes

Reviewer #2: Partly

3. Has the statistical analysis been performed appropriately and rigorously? 

Reviewer #1: Yes

Reviewer #2: N/A

4. Have the authors made all data underlying the findings in their manuscript fully available?

Reviewer #1: Yes

Reviewer #2: Yes

5. Is the manuscript presented in an intelligible fashion and written in standard English?

Reviewer #1: Yes

Reviewer #2: Yes

6. Review Comments to the Author

Reviewer #1: Many thanks for the opportunity to review the revised article. This remains good research, fit for publication, and I believe the authors have mostly addressed the issues discussed during the prior round of review. I offer 1 major issue and some additional brief comments and replies to the author’s responses. Assuming these are addressed, I would recommend for publication.

Major Issues:

My biggest remaining issue is the “Qualitative evaluation.” I thank the authors for their explanation of mixed-methods research but to claim you used a “mixed-methods methodology” that includes a “Qualitative” section you need to actually include and describe some sort of qualitative method used to evaluate the work. e.g., thematic analysis, grounded theory, content analysis, framework analysis etc. These are all well established, detailed, systematic methods for conducting qualitative analysis. A one sentence qualitative methods section that just states that the research was “assessed qualitatively” is not sufficient. The Cochrane Risk of Bias tool may have provided a deductive framework around which to categorise and report these evaluations (using something like content analysis) is what I was getting at. You should remove the word “Qualitative” from this section and replace it with something like “Narrative” pr “Subjective” assessments so as to not give the impression that a robust, systematic Qualitative method was used for this evaluation. At least, based on what is reported in the paper, that does not seem to be the case.

Minor Issues:

Many thanks to the authors for addressing how they selected the journals. They may be interested to see, if they have not already, that some very similar research was presented recently at the European Society of Cardiology Congress 2020. https://www.tctmd.com/news/covid-19-blamed-weaker-research-published-top-tier-journals-2020 The fact that they looked at the same three journals, however using different criteria/scales and historic controls. This is an interesting check on the findings from Zdravkovic et al. and can be brought out in the discussion. I couldn’t locate a book of abstracts for that conference (it may not be available yet) but the results are described in that article. Obviously not ideal for referencing, but the similarity is notable and probably worth being mentioned in relation to this research.

I am glad to see the sensitivity analysis removing editorials. I disagree that ~8% and ~14% of your sample is very small. That said it is very good that your findings remained robust to removing these. I think that your findings remained robust to the various sensitivity analyses is a strength.

I don’t think it is correct to say that the JAMA Research Letter format (or other journal’s similar formats) are indicative of the editors/reviewers believing the articles “are not good enough to meet the criteria for Original article category.” You can submit directly in the Research Letter format without being referred there by the editors. Some research simply doesn’t need 2000+ words to get the point across. That said, I think the inclusion of the sensitivity analysis is sufficient.

On the citation analysis, I agree that with the new data, it would be unreasonable to check all of these citations for context. However I do think the levels originally reported through May could reasonably have been investigated. I understand that you feel this is out of the scope of this paper and resources of your group and respect the decision not to investigate further at this time. I would simply request that if you agree that what I stated about citation context may be true, that it is mentioned as relevant to the interpretation of this finding on Page 14 and potentially as a direction for future research in the Discussion. A case study in this that has personally annoyed me quite a bit is Didier Raoult going on about how many times his hydroxychloroquine paper has been cited, as a defense of the paper, with no context of how many of those were citing it in the context of pointing out the many limitations of that research. No need to use this example but I think it proves the point.

I would like to applaud the authors for making their data openly available.

Many thanks again for the opportunity to review this paper. If these issues can be rectified to the editor’s satisfaction I am happy to recommend this paper go forward to publication.

Reviewer #2: 1. I would like to comment on the following author’s answer:

A-2-3: We fully agree with the reviewer. We had planed an observational study design as a case control. We were then obliged by PLOS ONE to include “systematic review” in the title, because we deal with study comparisons. We would like to follow your suggestion to go as a case-control study which was the initial plan, unless the editorial office overrules us.

Actually, I agree that this is not a systematic review therefore I would avoid any reference to this study design in order to not arise misunderstanding regarding the study type. I did not find the term “systematic review in the title accordingly and I agree with the removal of any reference to the PRISMA statement.

2. My major concern is still related to the quality appraisal the authors performed. I went through the references the authors reported to support the choice of QUALSYST tool use, the mixed methodology and still some issues and confusion raised.

I agree with their comment A1-6 B where the authors described the definition of qualitative and quantitative research but I think something is not still clear or there is some confusion in the description of this concept. They stated: Qualitative analysis is the analysis of qualitative data such as text data from interview transcripts. Unlike quantitative analysis, which is statistics-driven and largely independent of the researcher, qualitative analysis is heavily dependent on the researcher’s analytic and integrative skills and personal knowledge of the social context where the data is collected. The emphasis in qualitative analysis is “sense making” or understanding a phenomenon, rather than predicting or explaining. This methodology is very frequently used in social sciences in combination with quantitative analysis, the so called mixed-methods methodology.

I am aware regarding the mixed-method methodology but I don’t think this is the case to adopt it: here the author should have assessed just the methodological quality of the quantitative research study design (ie. experimental, observational etc….) offering a qualitative assessment and not a quantitative one which in the Cochrane wording is referring to a quantitative synthesis/statistically driven. Moreover, the second part the authors assessed, ie. “qualitative analysis” is just an assessment of reporting characteristic of the included studies - I did not assess it as a qualitative analysis. PLease look at the following examples of reporting characteristics assessment/methdoological quality: https://journals.plos.org/plosmedicine/article?id=10.1371/journal.pmed.0040078;
https://www.sciencedirect.com/science/article/pii/S0895435616308162.

Otherwise the study design is not clearly described – the mixed methodology might describe the study design and not the appraisal of the studies. Here an example: https://pubmed.ncbi.nlm.nih.gov/33033951/ where both a questionnaire(quantitative analysis) and focused groups (qualitative analysis) were performed.

In this case control study, this mixed methodology did not really reflect what was done. To my opinion is a control study belonging to the quantitative research world where included studies were compared using the following instruments:

1. Methodological quality throughout the QUALSYST tool (even though the high standard: Cochrane Risk of bias or the New Castle or the ROBINS I could have been used too)

2. Asssessment of reporting elements such as: reported from page 7, lines 150: ”The COVID-19 original research articles (n = 13) were assessed qualitatively to report on their major weaknesses (which type of weakness and how this is standardized across studies? Was not already included in the methodological quality for quantitative studies in the QUALSYST tool?), potential conflicts of interest, and likely influence on further research and clinical practice (in which way these are standardized, collected and reported in the assessment? Are regression analyses planned to investigate the influence? ).

The second point cannot be equal to a qualitative part of a mixed methodology where usually focus groups or interviews are used to collect qualitative data. The data/info the authors wanted to comment on are included and reported in the selected studies, in the manuscript/full text as general characteristics (ie. conflict of interest) and so are simply collected from them and then discussed.

I suggest the authors to better revise the study design performed, the qualitative/quantitative wording.

7. PLOS authors have the option to publish the peer review history of their article (what does this mean?). If published, this will include your full peer review and any attached files.

Reviewer #1: **Yes: **Nicholas J. DeVito

Reviewer #2: No

---

## [Author Response · Author response to Decision Letter 1]

20 Oct 2020

Reply to Reviewers for PONE-D-20-14688R1: 

Scientific Quality of COVID-19 and SARS CoV-2 Publications in the Highest Impact Medical Journals during the Early Phase of the Pandemic: A Case Control Study

Review Comments to the Author

Reviewer #1: Many thanks for the opportunity to review the revised article. This remains good research, fit for publication, and I believe the authors have mostly addressed the issues discussed during the prior round of review. I offer 1 major issue and some additional brief comments and replies to the author’s responses. Assuming these are addressed, I would recommend for publication.

Our reply: We thank the reviewer for the appreciation of our work

Major Issues:

My biggest remaining issue is the “Qualitative evaluation.” I thank the authors for their explanation of mixed-methods research but to claim you used a “mixed-methods methodology” that includes a “Qualitative” section you need to actually include and describe some sort of qualitative method used to evaluate the work. e.g., thematic analysis, grounded theory, content analysis, framework analysis etc. These are all well established, detailed, systematic methods for conducting qualitative analysis. A one sentence qualitative methods section that just states that the research was “assessed qualitatively” is not sufficient. The Cochrane Risk of Bias tool may have provided a deductive framework around which to categorise and report these evaluations (using something like content analysis) is what I was getting at. You should remove the word “Qualitative” from this section and replace it with something like “Narrative” pr “Subjective” assessments so as to not give the impression that a robust, systematic Qualitative method was used for this evaluation. At least, based on what is reported in the paper, that does not seem to be the case.

Our reply: Thank you for this comment. We have changed the wording “qualitative” to “narrative” throughout. 

Minor Issues:

Many thanks to the authors for addressing how they selected the journals. They may be interested to see, if they have not already, that some very similar research was presented recently at the European Society of Cardiology Congress 2020. https://www.tctmd.com/news/covid-19-blamed-weaker-research-published-top-tier-journals-2020 The fact that they looked at the same three journals, however using different criteria/scales and historic controls. This is an interesting check on the findings from Zdravkovic et al. and can be brought out in the discussion. I couldn’t locate a book of abstracts for that conference (it may not be available yet) but the results are described in that article. Obviously not ideal for referencing, but the similarity is notable and probably worth being mentioned in relation to this research.

Our reply: This is indeed very interesting. Thank you for making us aware. We have contacted Professor Stefanini and he provided us the slideset that he presented at the congress. He states that besides that, his data do not exist in a published and citable form yet. We have included a statement in the discussion. If this does not suit your citation policy, please advise us.

“One might argue that a comparison to a historical control group, for example the same time frame in 2019, when there was no pandemic effect on research, would have been more appropriate. Our hypothesis was that COVID-related research showed lower quality than non-COVID research. A historical control group may introduce a selection bias, since conditions for research then would be clearly different. We would therefore argue that the control group has to be subject to the same conditions as the test group, when methodological quality is assessed. This may be different for other endpoints like total research output. In line with our results, Stefanini et al reported - in an oral presentation at the European Society of Cardiology Congress 2020 - similar findings of lower quality associated with COVID-19 in the same journals and time frame as our work with a historical control group of 2019. So, both historical and contemporary control groups lead to the same conclusions.”

I am glad to see the sensitivity analysis removing editorials. I disagree that ~8% and ~14% of your sample is very small. That said it is very good that your findings remained robust to removing these. I think that your findings remained robust to the various sensitivity analyses is a strength.

Our reply: Thank you

I don’t think it is correct to say that the JAMA Research Letter format (or other journal’s similar formats) are indicative of the editors/reviewers believing the articles “are not good enough to meet the criteria for Original article category.” You can submit directly in the Research Letter format without being referred there by the editors. Some research simply doesn’t need 2000+ words to get the point across. That said, I think the inclusion of the sensitivity analysis is sufficient.

Our reply: Thank you

On the citation analysis, I agree that with the new data, it would be unreasonable to check all of these citations for context. However I do think the levels originally reported through May could reasonably have been investigated. I understand that you feel this is out of the scope of this paper and resources of your group and respect the decision not to investigate further at this time. I would simply request that if you agree that what I stated about citation context may be true, that it is mentioned as relevant to the interpretation of this finding on Page 14 and potentially as a direction for future research in the Discussion. A case study in this that has personally annoyed me quite a bit is Didier Raoult going on about how many times his hydroxychloroquine paper has been cited, as a defense of the paper, with no context of how many of those were citing it in the context of pointing out the many limitations of that research. No need to use this example but I think it proves the point.

Our reply: We fully agree with your notion and have expanded the discussion:

It must be acknowledged though, that a citation does not necessarily need to be positive for a study or author, if the context, i. e. criticisim or discussions about retractions and corrections, of the citations are considered. This is beyond the scope of our work.

I would like to applaud the authors for making their data openly available.

Our reply: Thank you

Many thanks again for the opportunity to review this paper. If these issues can be rectified to the editor’s satisfaction I am happy to recommend this paper go forward to publication.

Our reply: We thank you for your constructive criticism and hope that we have now sufficiently addressed all the points raised.

 

Reviewer #2: 1. I would like to comment on the following author’s answer:

A-2-3: We fully agree with the reviewer. We had planed an observational study design as a case control. We were then obliged by PLOS ONE to include “systematic review” in the title, because we deal with study comparisons. We would like to follow your suggestion to go as a case-control study which was the initial plan, unless the editorial office overrules us.

Actually, I agree that this is not a systematic review therefore I would avoid any reference to this study design in order to not arise misunderstanding regarding the study type. I did not find the term “systematic review in the title accordingly and I agree with the removal of any reference to the PRISMA statement.

Our reply: Thank you for supporting our idea. We have already changed the title according to your suggestion and journal editors seem to agree.

2. My major concern is still related to the quality appraisal the authors performed. I went through the references the authors reported to support the choice of QUALSYST tool use, the mixed methodology and still some issues and confusion raised.

I agree with their comment A1-6 B where the authors described the definition of qualitative and quantitative research but I think something is not still clear or there is some confusion in the description of this concept. They stated: Qualitative analysis is the analysis of qualitative data such as text data from interview transcripts. Unlike quantitative analysis, which is statistics-driven and largely independent of the researcher, qualitative analysis is heavily dependent on the researcher’s analytic and integrative skills and personal knowledge of the social context where the data is collected. The emphasis in qualitative analysis is “sense making” or understanding a phenomenon, rather than predicting or explaining. This methodology is very frequently used in social sciences in combination with quantitative analysis, the so called mixed-methods methodology.

I am aware regarding the mixed-method methodology but I don’t think this is the case to adopt it: here the author should have assessed just the methodological quality of the quantitative research study design (ie. experimental, observational etc….) offering a qualitative assessment and not a quantitative one which in the Cochrane wording is referring to a quantitative synthesis/statistically driven. Moreover, the second part the authors assessed, ie. “qualitative analysis” is just an assessment of reporting characteristic of the included studies - I did not assess it as a qualitative analysis. PLease look at the following examples of reporting characteristics assessment/methdoological quality: https://journals.plos.org/plosmedicine/article?id=10.1371/journal.pmed.0040078;
https://www.sciencedirect.com/science/article/pii/S0895435616308162.

Otherwise the study design is not clearly described – the mixed methodology might describe the study design and not the appraisal of the studies. Here an example: https://pubmed.ncbi.nlm.nih.gov/33033951/ where both a questionnaire(quantitative analysis) and focused groups (qualitative analysis) were performed.

In this case control study, this mixed methodology did not really reflect what was done. To my opinion is a control study belonging to the quantitative research world where included studies were compared using the following instruments:

1. Methodological quality throughout the QUALSYST tool (even though the high standard: Cochrane Risk of bias or the New Castle or the ROBINS I could have been used too)

2. Asssessment of reporting elements such as: reported from page 7, lines 150: ”The COVID-19 original research articles (n = 13) were assessed qualitatively to report on their major weaknesses (which type of weakness and how this is standardized across studies? Was not already included in the methodological quality for quantitative studies in the QUALSYST tool?), potential conflicts of interest, and likely influence on further research and clinical practice (in which way these are standardized, collected and reported in the assessment? Are regression analyses planned to investigate the influence? ).

The second point cannot be equal to a qualitative part of a mixed methodology where usually focus groups or interviews are used to collect qualitative data. The data/info the authors wanted to comment on are included and reported in the selected studies, in the manuscript/full text as general characteristics (ie. conflict of interest) and so are simply collected from them and then discussed.

I suggest the authors to better revise the study design performed, the qualitative/quantitative wording.

Our reply: Thank you. As suggested by reviewer 1, we changed the wording “qualitative” to “narrative” throughout the manuscript. We also changed the “Mixed-methods” wording to “multi-step”, to more accurately reflect the methodology used.

---

## [Editor Report · Decision Letter 2]

22 Oct 2020

Scientific Quality of COVID-19 and SARS CoV-2 Publications in the Highest Impact Medical Journals during the Early Phase of the Pandemic: A Case Control Study

PONE-D-20-14688R2

Dear Dr. Berger,

We’re pleased to inform you that your manuscript has been judged scientifically suitable for publication and will be formally accepted for publication once it meets all outstanding technical requirements.

Kind regards,

Bart Ferket

Academic Editor

PLOS ONE
---

## [Editor Report · Acceptance letter]

26 Oct 2020

PONE-D-20-14688R2 

Scientific Quality of COVID-19 and SARS CoV-2 Publications in the Highest Impact Medical Journals during the Early Phase of the Pandemic: A Case Control Study 

Dear Dr. Berger:

I'm pleased to inform you that your manuscript has been deemed suitable for publication in PLOS ONE. Congratulations! Your manuscript is now with our production department. 

Kind regards, 

on behalf of

Dr. Bart Ferket 

Academic Editor

PLOS ONE